# GUDA: Counterfactual Group-wise Training Data Attribution for Diffusion Models via Unlearning

**Naoki Murata** [1]  **Yuhta Takida** [1]  **Chieh-Hsin Lai** [1]  **Toshimitsu Uesaka** [1]  **Bac Nguyen** [1]  **Stefano Ermon** [2]
**Yuki Mitsufuji** [1 3]

## Abstract

Training-data attribution for vision generative models aims to identify which training data influenced a given output. While most methods score individual examples, practitioners often need group-level answers (e.g., artistic styles or object classes). Group-wise attribution is counterfactual: how would a model's behavior on a generated sample change if a group were absent from training? A natural realization of this counterfactual is Leave-One-Group-Out (LOGO) retraining, which retrains the model with each group removed; however, it becomes computationally prohibitive as the number of groups grows. We propose GUDA (Group Unlearning-based Data Attribution) for diffusion models, which approximates each counterfactual model by applying machine unlearning to a shared full-data model instead of training from scratch. GUDA quantifies group influence using differences in a likelihood-based scoring rule (ELBO) between the full model and each unlearned counterfactual. Experiments on CIFAR-10 and artistic style attribution with Stable Diffusion show that GUDA identifies primary contributing groups more reliably than semantic similarity, gradient-based attribution, and instance-level unlearning approaches, while achieving ∼100× speedup on CIFAR-10 over LOGO retraining. The code is available at https://github.com/sony/guda.

## 1. Introduction

Consider an AI-generated artwork that blends multiple artistic styles (Ho et al., 2020; Rombach et al., 2022). A fundamental question arises: which training data groups contributed most to this output? This question is critical for copyright assessment, fair compensation for data providers, and debugging (Carlini et al., 2023; Ghorbani & Zou, 2019; Henderson et al., 2023). We address the core problem of group-wise data attribution: estimating the counterfactual influence of training data groups on generated outputs.

**Instance-level vs. group-level attribution.** While instance-level training data attribution methods (Koh & Liang, 2017; Pruthi et al., 2020; Park et al., 2023; Dai & Gifford, 2023; Lin et al., 2025; Zheng et al., 2024) estimate which specific training samples influenced a generation, group-level attribution asks a fundamentally different question: how would the generated output change if an entire group (e.g., an artistic style, object class) were absent from training? Simply summing instance-level scores fails due to (i) *scalability*: evaluating per-instance contributions becomes prohibitive as datasets grow, and (ii) *nonlinearity*: group influences interact through shared representations, so collective effects are generally non-additive (Hu et al., 2024; Basu et al., 2020; Koh et al., 2019).

**Counterfactual models as the estimation target.** The natural gold standard for group attribution is the counterfactual model retrained without the target group. The quantity we seek to estimate is the difference in model behavior between the full-data model $\theta^{\text{full}}$ and the counterfactual model $\theta^{\text{logo}}_{-k}$ trained without group $k$, for which Leave-One-Group-Out (LOGO) retraining provides an oracle implementation. We measure group attribution by comparing the model's likelihood-based explanatory power (via ELBO as a tractable surrogate) on a sample under the full-data model versus the counterfactual model. However, LOGO retraining is prohibitively expensive: training $N + 1$ models from scratch (where $N$ is the number of groups) scales poorly as the number of groups increases, making it impractical when applied to large-scale models.

**Unlearning as a natural approximation.** Our key insight is that group attribution inherently asks for the effect of *data removal*: the counterfactual question "what if group $k$ were absent from training?" is precisely a data removal query. This is exactly the operation that machine unlearning ad-

[1]Sony AI, Tokyo, Japan [2]Stanford University, Stanford, CA, USA [3]Sony Group Corporation, Tokyo, Japan. Correspondence to: Naoki Murata <naoki.murata@sony.com>.

*Proceedings of the 43rd International Conference on Machine Learning*, Seoul, South Korea. PMLR 306, 2026. Copyright 2026 by the author(s).

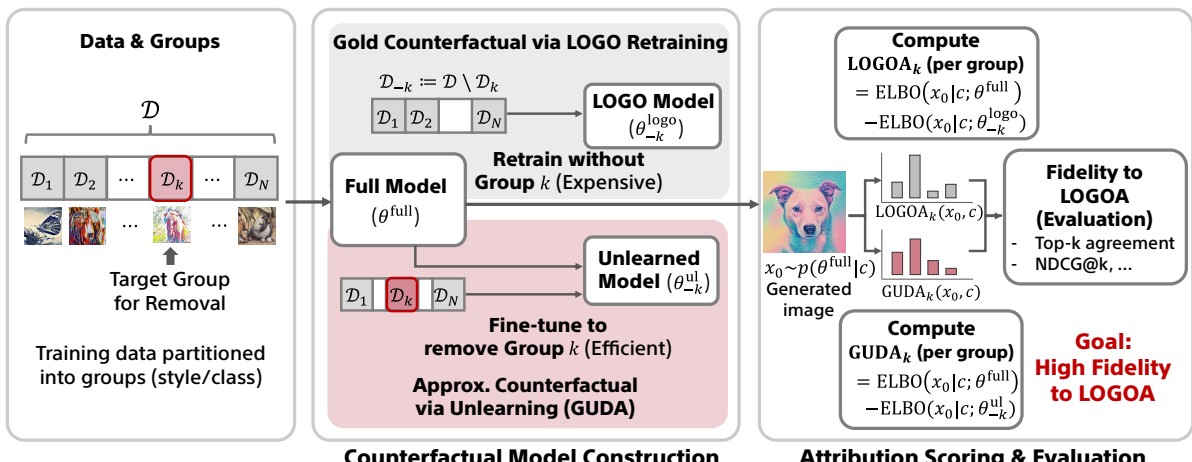

*Figure 1.* Overview of the GUDA (Group Unlearning-based Data Attribution) framework. Instead of retraining $N + 1$ models from scratch (LOGO), GUDA starts from the all-group model $\theta^{\text{full}}$ and applies machine unlearning to obtain approximate counterfactual models $\theta^{\text{ul}}_{-k}$ for each group $k$. Attribution is computed by comparing the ELBO of generated samples under the all-group model versus each unlearned model, providing an efficient approximation to the counterfactual target validated against LOGO.

dresses (Ginart et al., 2019; Bourtoule et al., 2021; Nguyen et al., 2025). This makes unlearning the *natural solution class* for approximating $\theta^{\text{logo}}_{-k}$: starting from $\theta^{\text{full}}$ and selectively removing a group's influence through fine-tuning, rather than expensive retraining from scratch.

Importantly, using unlearning *for attribution* is not a plug-in choice: deletion-oriented objectives do not target an explicit LOGO counterfactual. We therefore define a likelihood-based LOGO estimand for *per-generation* group influence and design redirection-based unlearning (including anchor conditioning) whose fidelity is validated against LOGO oracles.

We propose GUDA as a general framework that uses unlearning to approximate counterfactual models and estimates group influence by comparing their likelihood-based measures (ELBO difference). To achieve this, the unlearning loss consists of two terms: a shared preservation term to prevent catastrophic forgetting, and a setting-specific forget term that varies by generation setting. For unconditional generation, we adopt ReTrack-based redirection (Shi et al., 2026) that trains toward importance-weighted targets from the retain set. For conditional text-to-image attribution, we instead redirect forget-condition responses toward retain-style anchors, yielding a conditional analogue of ReTrack-style redirection. We summarize the overall GUDA pipeline in Fig. 1.

This paper makes the following contributions:

- **Framework:** We formalize per-generation group attribution in diffusion models around a LOGO-style counterfactual estimand, and provide an unlearning-based approximation that makes it practical at scale.

GUDA defines group influence as an ELBO difference between the full-data model and the counterfactual model $\theta^{\text{logo}}_{-k}$ (Sec. 3.3), and approximates $\theta^{\text{logo}}_{-k}$ by machine unlearning from a shared initialization rather than expensive from-scratch retraining (Sec. 4), with approximation quality validated against LOGO in experiments.

- **Two instantiations:** We demonstrate GUDA in two settings: (i) unconditional generation via ReTrack-based importance-weighted unlearning (Sec. 4.2.1), and (ii) conditional text-to-image style attribution via CLIP-weighted style-selection anchors for redirection (Sec. 4.2.2) for a conditional analogue of ReTrack.

- **Empirical validation:** We demonstrate on CIFAR-10 and artistic style attribution (Sec. 5) that GUDA substantially outperforms semantic similarity baselines (CLIPA) and instance-level attribution (Lin et al., 2025; Wang et al., 2024) in identifying the most influential groups, validating that directly targeting group-removal counterfactual effects is more effective than similarity-based or instance-level approximations.

**Conflict of Interest Disclosure.** The authors declare no conflicts of interest.

## 2. Related Work

### 2.1. Instance-Level Attribution and Group Effects

Influence functions (Koh & Liang, 2017), TracIn (Pruthi et al., 2020), and TRAK (Park et al., 2023) provide instance-level attribution with improved scalability. However, extending these to group effects poses challenges: Koh et al.

(2019) show that aggregating instance-level influences often fails to capture nonlinear group interactions, while Basu et al. (2020) address this through second-order approximations. From a data valuation perspective, Data Shapley (Ghorbani & Zou, 2019) and CS-Shapley (Schoch et al., 2022) provide principled frameworks based on cooperative game theory, and Lu et al. (2025) apply Shapley-style crediting to diffusion models for *global* contributor valuation using pruning/fine-tuning to accelerate subset evaluation. Unlike these approaches, which target specific value functions without constructing the counterfactual model $\theta_{-k}^{\text{logo}}$, GUDA targets this oracle directly. Datamodels (Ilyas et al., 2022) similarly predict model outputs as linear functions of training subsets without constructing $\theta_{-k}^{\text{logo}}$, representing a complementary predictive direction.

## 2.2. Data Attribution for Diffusion Models

Recent methods address attribution specifically for diffusion models: Dai & Gifford (2023), Lin et al. (2025), and Zheng et al. (2024) develop instance-level attribution techniques for these settings. Wang et al. (2023) establish evaluation protocols, and Wang et al. (2024) connect unlearning to attribution for text-to-image models. Our work differs from Wang et al. (2024) in two key aspects: (i) *Estimand*: we approximate the group-removal counterfactual model $\theta_{-k}^{\text{logo}}$, not instance-level attribution for a synthesized output; (ii) *Method*: we unlearn the target group to approximate $\theta_{-k}^{\text{logo}}$, rather than unlearning a synthesized image to obtain loss-based ranking signals. Choi et al. (2025) apply unlearning-based attribution to text-to-music models, targeting instance-level attribution in the music domain; this is a complementary direction to GUDA's group-level counterfactual attribution for vision diffusion models.

## 2.3. Machine Unlearning for Diffusion Models

Machine unlearning removes training data influence without full retraining, with foundational work by Ginart et al. (2019) and Bourtoule et al. (2021). For diffusion models, representative approaches include ESD (Gandikota et al., 2023), Forget-Me-Not (Zhang et al., 2024a), EraseDiff (Wu et al., 2025), and ReTrack (Shi et al., 2026). While these methods focus on concept or data removal as their primary objective, GUDA uses unlearning as a computational approximation of the LOGO counterfactual, not as a deletion mechanism. In particular, GUDA-U adopts ReTrack as its unlearning operator; the contribution is the LOGOA estimand and the LOGO-oracle-validated framework, not a novel unlearning algorithm.

**Novelty summary.** Traditional attribution methods return instance-level influences without constructing counterfactual models; unlearning methods modify model behavior without targeting an explicit attribution estimand. GUDA

bridges these directions through three contributions: (i) LOGOA, an explicit per-generation group-removal estimand with LOGO retraining as the oracle; (ii) unlearning-based approximation of each counterfactual model, validated directly against that oracle; and (iii) conditional anchor design that extends group-removal counterfactuals to text-to-image models where removing a style group shifts both image and prompt distributions.

## 3. Preliminaries

We formalize the group attribution problem, define our estimation target (counterfactual model via LOGO retraining), and introduce baseline approaches.

### 3.1. Setup: Group-wise Counterfactual Attribution

Let $\mathcal{D} := \bigcup_{k=1}^{N} \mathcal{D}_k$ be a training dataset, where $\{\mathcal{D}_k\}_{k=1}^{N}$ forms a partition into $N$ disjoint groups[1], and $\mathcal{D}_k$ denotes a semantic group (e.g., a CIFAR-10 class or an artistic style). We define the retain set excluding group $k$ as $\mathcal{D}_{-k} := \mathcal{D} \setminus \mathcal{D}_k$. Let $\theta^{\text{full}}$ be a generative model trained on $\mathcal{D}$, and let $x_0$ denote a *generated sample* for which we wish to compute attribution, where $c$ is the conditioning signal (for unconditional models, set $c = \varnothing$).

**Counterfactual definition of group influence.** Group attribution asks: *How would the model's behavior change if group $k$ had been absent during training?* This is naturally formalized by comparing the full-data model to a model trained on the retain set $\mathcal{D} \setminus \mathcal{D}_k$. We use "counterfactual" in an operational model-level sense, comparing models trained with and without a group, rather than as a unit-level causal counterfactual in a structural causal model. Group-wise attribution methods therefore target a *group-removal* counterfactual model, either explicitly (via retraining) or approximately (via unlearning).

**Measuring model behavior via likelihood.** We quantify *model behavior* on a generated sample $(x_0, c)$ by a scalar score that measures how well the model explains the observation. Ideally, we would use the log score $\log p_\theta(x_0|c)$ as a proper scoring rule. In this work, we will instantiate this score with a tractable surrogate for diffusion models (Sec. 3.2).

### 3.2. Diffusion Models and ELBO as a Likelihood Surrogate

We briefly recap DDPM-style diffusion models (Ho et al., 2020) to define the ELBO used throughout. We use $(x_0, x_t)$ to denote data and its noised version across both pixel-space

---

[1] Extending to overlapping groups, where a sample may belong to multiple groups, is an important practical direction left for future work.

and latent diffusion models; for Latent Diffusion Models (e.g., Stable Diffusion) (Rombach et al., 2022), $x_0$ corresponds to VAE latent representations.

A diffusion model is a latent-variable model with latent chain $x_{1:T}$ and data $x_0$. The forward (diffusion) process is fixed:

$$q(x_{1:T}|x_0) = \prod_{t=1}^{T} q(x_t|x_{t-1}), \quad (1)$$

where $q(x_t|x_{t-1}) = \mathcal{N}(\sqrt{1-\beta_t}\, x_{t-1},\, \beta_t I)$ for a variance schedule $\{\beta_t\}_{t=1}^{T}$. We also use the marginal $q_t(x_t|x_0) = \mathcal{N}(\sqrt{\bar{\alpha}_t}\, x_0, \sigma_t^2 I)$, where $\alpha_t := 1 - \beta_t$, $\bar{\alpha}_t := \prod_{s=1}^{t} \alpha_s$, and $\sigma_t := \sqrt{1 - \bar{\alpha}_t}$. The generative model learns a reverse process $p_\theta(x_{t-1}|x_t)$ with a prior $p(x_T)$:

$$p_\theta(x_{0:T}) = p(x_T) \prod_{t=1}^{T} p_\theta(x_{t-1}|x_t). \quad (2)$$

**ELBO and its relation to log-likelihood.** The marginal log-likelihood satisfies

$$\log p_\theta(x_0) \geq \mathbb{E}_{q(x_{1:T}|x_0)} \left[ \log p_\theta(x_{0:T}) - \log q(x_{1:T}|x_0) \right]$$
$$=: \text{ELBO}(x_0; \theta). \quad (3)$$

Therefore, a higher $\text{ELBO}(x_0; \theta)$ implies that the model assigns a higher (lower-bounded) likelihood to $x_0$. For conditional diffusion models (e.g., text-to-image), we analogously define $\text{ELBO}(x_0|c; \theta)$ where $c$ is the conditioning signal, providing a lower bound on $\log p_\theta(x_0|c)$. In practice, DDPM training optimizes a variational bound objective, and common noise-prediction losses correspond to a weighted form of this bound.

### 3.3. LOGOA: Scoring Difference as Counterfactual Group Influence

We use LOGO (Leave-One-Group-Out) to refer to the retraining procedure, and LOGOA (LOGO Attribution) to refer to the resulting attribution score.

**Defining counterfactual influence.** Group attribution targets a counterfactual quantity: the difference in a model's ability to explain a sample $(x_0, c)$ between the full-data model and the counterfactual model without group $k$. Ideally, we would measure this using log-likelihood, which quantifies how well a probabilistic model explains an observed sample.

**ELBO as a tractable proxy.** While log-likelihood $\log p_\theta$ can be computed via probability flow ODE (Song et al., 2021), it is prohibitively expensive for repeated evaluation. We therefore use the ELBO as a tractable lower bound on log-likelihood (Eq. (3)). While ELBO does not guarantee preservation of likelihood ordering, it provides a practical surrogate commonly adopted for comparing generative models, including likelihood-based diffusion classifiers (Li

et al., 2023; Clark & Jaini, 2023). We empirically validate this proxy by comparing $\Delta\text{ELBO}$ against $\Delta \log p$ obtained via probability-flow ODE estimation on CIFAR-10 (Appendix D.5): the two are strongly correlated, and while KL-gap asymmetry between the full and counterfactual models can in principle distort lower-ranked groups, head identification remains stable. We therefore define LOGOA using ELBO differences:

$$\text{LOGOA}_k(x_0, c) = \text{ELBO}(x_0|c; \theta^{\text{full}})$$
$$- \text{ELBO}(x_0|c; \theta_{-k}^{\text{logo}}), \quad (4)$$

where $\theta_{-k}^{\text{logo}}$ is the counterfactual model obtained via retraining on $\mathcal{D}_{-k}$. A positive score means the full model explains $(x_0, c)$ better than the counterfactual model without group $k$, indicating that group $k$'s training data contributed to the generation of $(x_0, c)$. Conversely, a negative or near-zero score suggests group $k$ had little influence.

**Remark (LOGO oracle in practice).** Throughout, $\theta_{-k}^{\text{logo}}$ denotes the conceptual counterfactual obtained by training on $\mathcal{D}_{-k}$. In large-scale diffusion settings, we instantiate this oracle by *retraining from a fixed pretrained initialization* with the same recipe and budget, i.e., a practical "fine-tuning LOGO" proxy, which we use consistently in Sec. 5.

### 3.4. Similarity-based Attribution (CLIPA) as Baseline and Its Limitations

A common proxy for attribution is semantic similarity between a generated sample and training groups. For images, CLIP embeddings (Radford et al., 2021) provide a convenient representation for assessing semantic similarity, enabling efficient retrieval.

**CLIPA baseline.** We define CLIP-based attribution (CLIPA) by averaging cosine similarity in CLIP embedding space to images in each group $\mathcal{D}_k$: $\text{CLIPA}_k(x_0) = \frac{1}{|\mathcal{D}_k|} \sum_{x' \in \mathcal{D}_k} \text{CLIP-sim}(x_0, x')$, where $\text{CLIP-sim}(\cdot, \cdot)$ denotes CLIP cosine similarity.

**Similarity versus explanatory power.** Unlike LOGOA's scoring rule difference, which measures how well the model explains the sample, similarity measures association in embedding space without capturing counterfactual effects of data removal (Park et al., 2023; Zheng et al., 2024). In our experiments, CLIPA can be competitive on rank-based metrics (e.g., NDCG@3), but is typically weaker on head-identification metrics such as Top-1.

## 4. Method: GUDA (Group Unlearning-based Data Attribution)

Having established the estimation target (LOGOA), we now present GUDA, a framework for efficiently approximating

**Algorithm 1** GUDA Framework for Group Attribution

---

1: **Input:** Full-data model $\theta^{\text{full}}$, group partition $\{\mathcal{D}_k\}_{k=1}^N$, query sample $(x_0, c)$ (set $c = \varnothing$ for unconditional), unlearning operator $\mathcal{U}$

2: **Output:** Group attribution scores $\{\text{GUDA}_k(x_0, c)\}_{k=1}^N$

3: // *Precompute counterfactual models (can be done once in advance)*

4: **for** each group $k = 1, \ldots, N$ **do**

5:     $\theta_{-k}^{\text{ul}} \leftarrow \mathcal{U}(\theta^{\text{full}}; \mathcal{D}_k, \mathcal{D}_{-k})$
    // Sec. 4.2.1 (GUDA-U) or Sec. 4.2.2 (GUDA-C)

6: **end for**

7: // *Compute scoring rule difference for query (Eq. (5))*

8: **for** each group $k = 1, \ldots, N$ **do**

9:     $\text{GUDA}_k(x_0, c)$
    $\leftarrow \text{ELBO}(x_0|c; \theta^{\text{full}}) - \text{ELBO}(x_0|c; \theta_{-k}^{\text{ul}})$

10: **end for**

11: **return** $\{\text{GUDA}_k(x_0, c)\}_{k=1}^N$

---

the counterfactual models $\theta_{-k}^{\text{logo}}$ via machine unlearning. The key idea is to start from the full-data model and remove a group's influence by unlearning, instead of retraining from scratch.

## 4.1. Overview: The GUDA Framework

Fig. 1 illustrates the end-to-end workflow, and Algorithm 1 provides the corresponding procedural description. The workflow has two stages: (i) construct an approximate counterfactual model for each group via unlearning, and (ii) compute attribution for a query sample by comparing ELBO under the full model and the corresponding unlearned model.

The unlearning operator $\mathcal{U}$ differs for unconditional versus text-conditional settings (detailed in Sec. 4.2.1–4.2.2), but the overall framework remains the same: iterate over groups, construct counterfactual models, and compute score differences. Here, $\theta_{-k}^{\text{ul}}$ serves as an efficient approximation to the oracle counterfactual model $\theta_{-k}^{\text{logo}}$ (Sec. 3.3), which would require expensive retraining from scratch. Importantly, the unlearning procedure can be performed *in advance*: once the approximated counterfactual models $\{\theta_{-k}^{\text{ul}}\}_{k=1}^N$ are constructed, attribution for any generated sample requires only evaluating the ELBO under each precomputed unlearned model, without re-running the unlearning process.

## 4.2. GUDA: Unlearning-based Approximation of Counterfactual Models

GUDA approximates the counterfactual model $\theta_{-k}^{\text{logo}}$ by applying machine unlearning to the full-data model $\theta^{\text{full}}$, rather than retraining from scratch.

**Attribution score.** Given an approximate counterfactual model $\theta_{-k}^{\text{ul}}$, GUDA measures group influence using the same scoring-rule difference as LOGOA:

$$\text{GUDA}_k(x_0, c) = \text{ELBO}(x_0|c; \theta^{\text{full}}) - \text{ELBO}(x_0|c; \theta_{-k}^{\text{ul}}). \tag{5}$$

The key question is how to efficiently obtain $\theta_{-k}^{\text{ul}}$ that approximates the counterfactual target $\theta_{-k}^{\text{logo}}$.

**Unlearning as counterfactual approximation.** Unlearning provides a natural tool for approximating $\theta_{-k}^{\text{logo}}$: starting from $\theta^{\text{full}}$ and selectively removing group $k$'s influence through fine-tuning. We denote the unlearning operator as $\theta_{-k}^{\text{ul}} = \mathcal{U}(\theta^{\text{full}}; \mathcal{D}_k, \mathcal{D}_{-k})$. Various unlearning methods have been proposed with diverse loss formulations (Gandikota et al., 2023; Wu et al., 2025; Shi et al., 2026). Following common practice in the literature, we structure the objective as:

$$\mathcal{L}_{\text{unlearn}} = \mathcal{L}_{\text{forget}} + \lambda_{\text{pres}} \mathcal{L}_{\text{preserve}}, \tag{6}$$

where $\mathcal{L}_{\text{forget}}$ removes group $k$'s influence and $\mathcal{L}_{\text{preserve}}$ maintains performance on the retain set; both terms are instantiated in Sec. 4.2.1–4.2.2. We emphasize that GUDA uses unlearning as a *computational approximation*, not as a certified deletion mechanism.

**Common preservation term.** For both unconditional and conditional settings, we adopt a score-matching preservation loss inspired by Chen et al. (2025):

$$\mathcal{L}_{\text{preserve}} = \mathbb{E}_{(x,c) \sim \mathcal{D}_{-k}, t, \varepsilon} \left[ \| \epsilon_\theta(x_t, t, c) - \epsilon_{\theta^{\text{full}}}(x_t, t, c) \|_2^2 \right], \tag{7}$$

where $\theta^{\text{full}}$ is frozen. This encourages the unlearned model to match the original model's score predictions on retain-set samples, preventing catastrophic forgetting. The key difference between settings lies in $\mathcal{L}_{\text{forget}}$, detailed below.

### 4.2.1. GUDA-U: UNCONDITIONAL GENERATION VIA RETRACK-BASED REDIRECTION

For unconditional diffusion, the counterfactual model corresponds to training on $\mathcal{D}_{-k}$ with the standard noise-prediction objective.

**Core idea: importance-weighted redirection.** ReTrack (Shi et al., 2026) provides a principled approach to approximate this objective. The key insight is that when we denoise a noisy forget-set sample $x_0^{(f)} \in \mathcal{D}_k$, we can *redirect* the denoising target toward nearby samples in $\mathcal{D}_{-k}$, weighted by their proximity under the forward diffusion process.

Specifically, for a noised latent $x_t$ derived from $x_0^{(f)}$, each retain-set sample $x_0^{(r)} \in \mathcal{D}_{-k}$ contributes with weight proportional to $q_t(x_t|x_0^{(r)})$, which represents the probability

that $x_t$ could have originated from $x_0^{(r)}$. The forget loss trains the model toward an importance-weighted target:

$$\mathcal{L}_{\text{forget}}^{(U)} = \mathbb{E}_{x_0^{(f)} \sim \mathcal{D}_k, t, x_t} \left[ \|\epsilon_\theta(x_t, t) - \bar{\varepsilon}_t(x_t)\|_2^2 \right], \quad (8)$$

where $\bar{\varepsilon}_t(x_t) = \sum_{x_0^{(r)} \in \mathcal{D}_{-k}} w_t(x_t; x_0^{(r)}) \cdot (x_t - \sqrt{\bar{\alpha}_t}\, x_0^{(r)})/\sigma_t$ with importance weights $w_t(x_t; x_0^{(r)}) \propto q_t(x_t|x_0^{(r)})$ normalized over $\mathcal{D}_{-k}$, and $(\bar{\alpha}_t, \sigma_t)$ are diffusion schedule parameters (see Appendix A.1).

**Theoretical justification and practical approximation.** The full ReTrack objective, which includes an explicit density-ratio correction (Eq. (15) in Appendix A.1), equals the retain-only training objective in expectation and thus targets the same counterfactual model $\theta_{-k}^{\text{logo}}$ as LOGO retraining. Our practical loss (Eq. (8)) omits the density ratio and truncates the importance-weighted sum to the $K$ nearest neighbors, yielding a computationally efficient approximation motivated by this equivalence; its fidelity is validated empirically against LOGO in Sec. 5.

### 4.2.2. GUDA-C: CONDITIONAL TEXT-TO-IMAGE VIA WEIGHTED STYLE-SELECTION ANCHORS

In conditional text-to-image diffusion, groups can be defined by artists, object categories, or visual concepts. We focus on *artistic styles*, which also enables LOGO-based oracle evaluation. Our counterfactual question is: *how would the model's conditional predictions change if style $k$ were absent from training?*

**Why a naive conditional extension of ReTrack fails.** ReTrack's justification in the unconditional setting is that the retain-only (LOGO) objective can be rewritten as a denoising problem, where the optimal target is the posterior mean over the retain set. This rewrite only changes *how we sample* $x_t$ while keeping the training domain unchanged, since there is no conditioning variable.

In the conditional setting, however, the LOGO target is $\mathcal{L}_{\text{ret}}^{(C)}(\theta) = \mathbb{E}_{(x_0,c)\sim\mathcal{D}_{-k},t,\varepsilon}[\|\epsilon_\theta(x_t,t,c)-\varepsilon\|_2^2]$, so removing style $k$ shifts not only the image distribution but also the *prompt* distribution. Prompts containing style $k$ (i.e., $c_f$) are out-of-support under retain-only training, meaning that a ReTrack-style posterior target conditioned on $c_f$ is not well-defined, a condition-distribution mismatch that the anchor construction below is designed to resolve.

Therefore, to construct a conditional analogue, we must (i) map the forget condition to an on-distribution retain-only condition, and (ii) keep it compatible with the current noisy latent $x_t$ from a forget image.

**Anchor construction (keep content, swap style).** Given a forget pair $(x_f, c_f) \sim \mathcal{D}_k$, we construct an anchor condition $c_a$ by keeping the content in $c_f$ fixed and replacing only the style description. Concretely, we sample a retain style $s \in \mathcal{S}_{\text{retain}}$ (excluding $k$) from a CLIP-similarity-weighted distribution $\pi_s(\cdot|c_f)$, then synthesize a prompt that combines the original content with descriptors from $s$ (Appendix B.1). For instance, redirecting Abstractionism → Artist Sketch replaces style descriptors: *dynamic forms, energetic, energetic composition → grayscale, sketchy, soft shading*. This yields anchors whose prompts lie in the retain prompt distribution, resolving the condition-distribution mismatch while remaining semantically compatible with the forget content.

**Forget loss via anchor redirection.** We train the unlearned model to match the frozen full model's prediction under the anchor condition:

$$\mathcal{L}_{\text{forget}}^{(C)} = \mathbb{E}_{(x_f,c_f)\sim\mathcal{D}_k, t, \varepsilon, c_a\sim\mathcal{A}_{\text{WSS}}(c_f)}$$
$$\left[ \|\epsilon_\theta(x_t, t, c_f) - \epsilon_{\theta^{\text{full}}}(x_t, t, c_a)\|_2^2 \right], \quad (9)$$

where $x_t = \sqrt{\bar{\alpha}_t}\, x_f + \sigma_t \varepsilon$, $\theta^{\text{full}}$ is frozen, and $\mathcal{A}_{\text{WSS}}$ denotes the weighted style-selection anchor. Specifically, $\mathcal{A}_{\text{WSS}}(c_f)$ samples a retain style $s \in \mathcal{S}_{\text{retain}}$ from the CLIP-weighted distribution $\pi_s(s|c_f)$ and synthesizes $c_a$ by keeping the content in $c_f$ while replacing only the style descriptors with those of $s$ (Appendix B.1, including ablation studies against simpler anchor alternatives).

### 4.3. Computational Cost Analysis

LOGOA requires training $N + 1$ models from scratch (one full-data model plus $N$ leave-one-group-out models), while GUDA trains the full-data model once and performs lightweight unlearning fine-tuning for each group. The unlearned models can be precomputed and reused: query-time evaluation requires only ELBO computation under each counterfactual model, without per-example bookkeeping. Wall-clock timing results are reported in Tab. 1 (CIFAR-10) and Tab. 2 (UnlearnCanvas).

## 5. Experiments

We validate our framework in two experiments. First, we conduct preliminary validation on CIFAR-10 to establish proof of concept for group-wise attribution. Second, we perform evaluation on artistic style attribution using the UnlearnCanvas dataset (Zhang et al., 2024b) with Stable Diffusion 1.5 (Rombach et al., 2022), demonstrating practical applicability to modern text-to-image models. In addition to attribution accuracy, we report wall-clock time for end-to-end cost (preprocessing plus query-time), which is critical for scaling to many groups. Full details are provided in the Appendix.

**Evaluation design.** Our primary goal is to identify the most influential groups (the head), the key practical use case. We therefore emphasize head-focused evaluation metrics (e.g., NDCG@3, MRR, Top-1), which measure whether methods

*Table 1.* Group-wise attribution results and wall-clock time on CIFAR-10 (10 classes, 2,048 queries). Best in **bold**; ↑ higher is better. Times in hours:minutes format (h:mm).

| Method | Top-1↑ | MRR↑ | NDCG@3↑ | Top-3↑ | RBO↑ | Spearman↑ | Preproc. | Query | Total |
|---|---|---|---|---|---|---|---|---|---|
| GUDA (Ours) | **0.727** | **0.798** | **0.677** | **0.475** | **0.423** | 0.265 | 1:07 | 0:55 | 2:02 |
| GUDA w/ ESD | 0.619 | 0.732 | 0.634 | 0.457 | 0.406 | 0.241 | 0:38 | 0:55 | 1:33 |
| CLIPA | 0.662 | 0.755 | 0.646 | 0.462 | 0.412 | 0.246 | <00:01 | <00:01 | <00:01 |
| DAS (Lin et al., 2025) | 0.716 | 0.794 | 0.675 | 0.473 | 0.422 | **0.267** | 28:47 | 6:36 | 35:24 |
| D-TRAK (Zheng et al., 2024) | 0.609 | 0.731 | 0.639 | 0.466 | 0.409 | 0.258 | 23:43 | 6:47 | 30:30 |
| TRAK (Park et al., 2023) | 0.118 | 0.313 | 0.317 | 0.313 | 0.309 | 0.030 | 24:15 | 6:43 | 30:58 |
| LOGOA (oracle) | | | — | | | | 206:52 | 0:55 | 207:47 |

*Table 2.* Group-wise attribution results and wall-clock time on UnlearnCanvas (16 styles, 320 queries). Models are trained/unlearned in a 60-style setting; attribution is evaluated over 16 target styles. Best in **bold**; ↑ higher is better. Times in hours:minutes format (h:mm).

| Method | Top-1↑ | MRR↑ | NDCG@3↑ | Top-3↑ | RBO↑ | Spearman↑ | Preproc. | Query | Total |
|---|---|---|---|---|---|---|---|---|---|
| GUDA (Ours) | **0.456** | **0.582** | **0.734** | **0.405** | **0.446** | **0.239** | 7:18 | 1:36 | 8:54 |
| CLIPA | 0.338 | 0.467 | 0.672 | 0.291 | 0.393 | 0.117 | <00:01 | <00:01 | <00:01 |
| Wang et al. (Wang et al., 2024) | 0.047 | 0.214 | 0.588 | 0.249 | 0.355 | 0.147 | 2:22 | 156:11 | 158:33 |
| LOGOA (oracle) | | | — | | | | 43:15 | 2:53 | 46:08 |

identify primary influences (definitions in Appendix C.3). We also report Spearman correlation as a full-ranking measure, though it weights all positions equally, including the less informative tail.

## 5.1. Proof of Concept for Group-wise Attribution on CIFAR-10

### 5.1.1. MOTIVATION

We first validate on CIFAR-10, whose 10 separable classes enable systematic comparison against LOGO without semantic overlap. This addresses whether unlearning-based approximation outperforms existing attribution methods when ground truth is well-defined.

### 5.1.2. SETUP

We trained an all-group model on CIFAR-10, 10 LOGO models excluding one class each, and 10 unlearned models using the ReTrack forget loss (Sec. 4.2.1) combined with the common score-matching preservation loss (Eq. (7)). For comparison, we also created unlearned models using the ESD forget loss (Gandikota et al., 2023) with the same preservation term, and evaluated gradient-based attribution methods (DAS (Lin et al., 2025), D-TRAK (Zheng et al., 2024), TRAK (Park et al., 2023)). For DAS, D-TRAK, and TRAK, we aggregated instance-level scores to obtain group-level attribution; ablations on aggregation strategies are provided in Appendix D.2. We generated 2,048 samples from $\theta^{\text{full}}$ and computed attribution scores for all methods.

### 5.1.3. RESULTS AND MAIN OBSERVATIONS

Table 1 presents the results. GUDA (ReTrack) achieves the best or tied-best head-focused metrics: **72.7% Top-1 agreement** and NDCG@3 of 0.677, while being substantially faster than the nearest competitor (DAS). Key observations include:

**(1) Unlearning method matters.** GUDA with the ReTrack forget loss outperforms GUDA with the ESD forget loss (72.7% versus 61.9% Top-1), validating our choice of the ReTrack forget loss based on its theoretical connection to LOGO retraining through importance sampling (Appendix A.1). More broadly, this gap shows that attribution fidelity within the GUDA framework is directly determined by how well the chosen unlearning operator approximates the LOGO counterfactual; future improvements to unlearning quality will translate directly to better attribution.

**(2) Semantic similarity is insufficient.** CLIPA achieves 66.2% Top-1 agreement, which is reasonable but substantially below GUDA. This gap demonstrates that similarity in embedding space does not fully capture group-removal counterfactual influence, even when classes are visually distinct.

**(3) Gradient-based methods struggle.** D-TRAK achieves only 60.9% Top-1 agreement, below GUDA and CLIPA. TRAK achieves near-random Top-1 accuracy (11.8% versus 10% chance), likely due to limitations in adapting influence functions to unconditional generative models.

**Computational efficiency.** Table 1 shows wall-clock time for the 2,048-query CIFAR-10 experiment. The key find-

ing is that GUDA achieves a $\sim 100\times$ wall-clock speedup over the LOGO oracle (2h 02m vs. 207h 47m total). This speedup arises because unlearning requires only $\sim 1/120$ of the optimization steps compared to full retraining from scratch (20 epochs vs. 2,400 epochs). Compared to gradient-based DAS, GUDA also offers faster query-time evaluation (1.6s vs. 11.6s per image), making it more practical for large-scale attribution tasks. Detailed timing breakdown is provided in Appendix D.3.

### 5.1.4. IMPLICATION

These results validate unlearning-based counterfactual approximation when groups are clearly separable. GUDA consistently outperforms both gradient-based methods and semantic similarity baselines on head metrics, demonstrating that directly targeting the group-removal counterfactual is more effective than aggregating instance-level attributions.

However, CIFAR-10 lacks two challenges inherent to practical text-to-image attribution: (i) *conditioning*, where attribution depends on the interaction between an image and its prompt, and (ii) *prompt shift*, where evaluation prompts differ from training prompts. To study these effects while retaining oracle-based evaluation, we next consider a text-to-image benchmark with controlled LOGO comparisons that introduces these challenges.

### 5.2. Artistic Style Attribution on UnlearnCanvas

#### 5.2.1. MOTIVATION AND SETUP

We use UnlearnCanvas (Zhang et al., 2024b) with Stable Diffusion 1.5, balancing two requirements: (i) it is structured enough to support LOGO-style oracle evaluation via shared object categories across styles, and (ii) it captures key practical aspects of text-to-image attribution, including multimodal conditioning and prompt distribution shift at inference time.

We fine-tuned SD 1.5 on all 60 artistic styles from UnlearnCanvas to obtain $\theta^{\text{full}}$. For computational feasibility, we evaluate attribution on the first 16 styles in alphabetical order. Each LOGO model $\theta^{\text{logo}}_{-k}$ is trained on 59 styles excluding target style $k$, and each unlearned model $\theta^{\text{ul}}_{-k}$ is obtained by unlearning style $k$ from $\theta^{\text{full}}$; both retain the remaining 59 styles. All attribution metrics are computed over this 16-style evaluation subset. Accordingly, for UnlearnCanvas, we construct LOGO oracles by fine-tuning from the same SD 1.5 checkpoint as $\theta^{\text{full}}$ (rather than training from scratch), reflecting standard practice and computational realism for large pretrained diffusion models.

To prevent reliance on superficial prompt cues, we do not use the prompts provided with UnlearnCanvas (e.g., "A Dogs image in Abstractionism style"). Instead, we construct a new prompt set for more practical evaluation (e.g., "A depiction of Dogs, artistic style featuring energetic composition, impasto, dynamic forms"). See Appendix C.2.3 for details.

#### 5.2.2. RESULTS

Table 2 presents the central result. GUDA achieves **45.6% Top-1 agreement**, outperforming CLIPA (33.8%) and Wang et al. (2024) (4.7%). Beyond exact Top-1 matches, GUDA more closely tracks the LOGOA head: it improves **Top-3 overlap to 0.405** and achieves the best **NDCG@3 (0.734)** and **RBO (0.446)** among all methods, indicating stronger agreement on the most influential styles. Notably, GUDA also achieves the highest Spearman correlation (0.239), suggesting improved full-ranking alignment with LOGOA beyond just head performance.

**Qualitative results.** Figure 2 shows two representative examples. GUDA and CLIPA detect relatively high-ranked styles, while Wang et al.'s method deviates from LOGOA's target style.

#### 5.2.3. WHY HEAD-FOCUSED METRICS MATTER

Full-ranking correlation can obscure failures on the head; for example, Wang et al. achieve higher Spearman correlation than CLIPA (0.147 vs. 0.117) despite near-random Top-1 agreement (4.7%), since Spearman weights all ranks equally and can reward tail agreement. Therefore, we emphasize Top-$k$ metrics as the primary evaluation criterion to reflect correctness on the most influential groups.

#### 5.2.4. DISCUSSION

These results reveal two key insights. First, semantic similarity does not fully capture group-removal counterfactual influence for head identification. While CLIPA achieves competitive performance on some rank-based metrics (e.g., NDCG@3), GUDA substantially outperforms it on head-identification metrics (Top-1, MRR, RBO). CLIPA measures visual similarity; GUDA measures counterfactual influence through data removal.

Second, instance-level attribution methods do not scale efficiently to group attribution, both conceptually and computationally. On CIFAR-10, gradient-based methods (D-TRAK, TRAK) underperformed even the similarity baseline. On UnlearnCanvas, Wang et al.'s approach, which unlearns a synthesized image to compute loss-based rankings, performs near randomly, suggesting that instance-level unlearning signals do not capture group-level counterfactual effects.

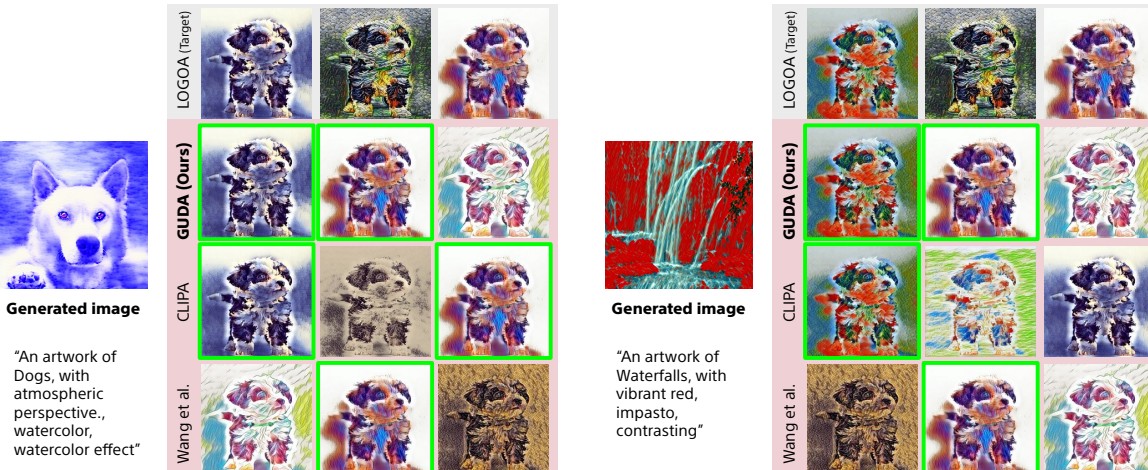

*Figure 2.* Qualitative comparison of group-wise attribution methods on UnlearnCanvas. For two generated images, we show the top-3 attributed styles for each method. Each row corresponds to: LOGOA (oracle target), GUDA (Ours), CLIPA, and Wang et al. Representative training images from each attributed style are displayed. Green boxes indicate agreement with the LOGOA top-3 styles.

## 6. Conclusion

We propose GUDA (Group Unlearning-based Data Attribution), a framework for group-wise data attribution that approximates expensive LOGO retraining through machine unlearning. Our experiments on CIFAR-10 and artistic style attribution demonstrate that GUDA achieves the best or tied-best head-focused metrics among all compared methods, including semantic similarity baselines and prior unlearning-based methods, while reducing wall-clock time by an order of magnitude compared to LOGO retraining and offering faster per-query evaluation than gradient-based alternatives (Tab. 1, 2). The framework adapts to different generation settings by choosing suitable unlearning objectives, opening avenues for scalable group-level understanding of generative models; as more LOGO-aligned unlearning methods emerge, they can be incorporated into GUDA with attribution quality verifiable against the LOGO oracle. Limitations of the current work are discussed in Appendix E.

## Acknowledgements

We are grateful to our colleagues Joan Serrà and Junghyun Koo for providing valuable feedback prior to submission.

## Impact Statement

This paper presents work whose goal is to advance the field of machine learning, specifically training data attribution for generative models.

**Positive Societal Impacts:** Our method enables more accurate identification of which training data groups contributed to generated outputs, which could support fair compensation systems for artists and data providers, improve model transparency, and facilitate debugging of problematic data sources.

**Potential Risks:** While our method focuses on group-level attribution, improved attribution techniques could potentially be extended to identify specific training examples, raising privacy considerations. Additionally, attribution methods could be used in content disputes, which may support legitimate copyright claims but could also lead to unintended legal consequences. The unlearning operator used in GUDA is not a certified deletion mechanism and provides no cryptographic or information-theoretic deletion guarantees; it should not be deployed in contexts requiring verifiable data removal. Our experiments focus on artistic styles and object classes; extending to other group definitions, particularly those involving demographic categories, would require careful consideration of fairness implications.

We believe the benefits for transparency and accountability outweigh the potential risks, but encourage responsible deployment.

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

# A. Theoretical Foundations

**Note on ELBO scope.** For latent diffusion models (e.g., Stable Diffusion), ELBO computation throughout this work is performed in latent space only and does not include the VAE decoder likelihood term $\log p(x|z)$. This is consistent with standard practice in latent diffusion (Rombach et al., 2022) and does not affect relative comparisons between models since the VAE decoder is shared and fixed across all models.

## A.1. ReTrack: Theoretical Foundation for GUDA-U

This appendix explains why ReTrack-style redirection is a theoretically aligned choice for approximating the LOGO counterfactual in the unconditional setting. The key point is that ReTrack is not an ad hoc "erase" objective: Before approximation, it can be derived as an *unbiased* reformulation of the retain-only diffusion training objective (i.e., the LOGO target), and its practical k-NN form follows from exponential weight decay.

**Setup and the LOGO target (retain-only training).** Fix a group $k$ to remove. Let $\mathcal{D}_{\text{ret}} := \mathcal{D}_{-k}$ and $\mathcal{D}_{\text{for}} := \mathcal{D}_k$. For an unconditional $\epsilon$-prediction diffusion model, the LOGO counterfactual $\theta_{-k}^{\text{logo}}$ corresponds to minimizing the standard diffusion loss over the retain set:

$$\theta_{-k}^{\text{logo}} \in \arg\min_{\theta} \ L_{\text{ret}}(\theta), \qquad L_{\text{ret}}(\theta) := \mathbb{E}_{x_0 \sim \mathcal{D}_{\text{ret}}, t, \varepsilon} \left[ \|\epsilon_{\theta}(x_t, t) - \varepsilon\|_2^2 \right], \tag{10}$$

where $x_t = \sqrt{\bar{\alpha}_t} x_0 + \sigma_t \varepsilon$ and $(\bar{\alpha}_t, \sigma_t)$ follow the forward diffusion schedule.

**An unbiased reformulation via importance sampling (ReTrack's core idea).** Define the *retain-mixture* density at timestep $t$:

$$p_{\text{ret},t}(x_t) := \frac{1}{|\mathcal{D}_{\text{ret}}|} \sum_{x_0^{(r)} \in \mathcal{D}_{\text{ret}}} q_t(x_t|x_0^{(r)}), \tag{11}$$

and analogously the *forget-mixture* density $p_{\text{for},t}(x_t) := \frac{1}{|\mathcal{D}_{\text{for}}|} \sum_{x_0^{(f)} \in \mathcal{D}_{\text{for}}} q_t(x_t|x_0^{(f)})$. Sampling $(x_0, t, \varepsilon)$ from the retain set in (10) is equivalent to sampling $x_t \sim p_{\text{ret},t}$ and then sampling the "source" retain example from the posterior

$$\pi_t(x_0^{(r)}|x_t) = \frac{q_t(x_t|x_0^{(r)})}{\sum_{x_0^{(r)'} \in \mathcal{D}_{\text{ret}}} q_t(x_t|x_0^{(r)'})} \quad \text{(uniform prior over } \mathcal{D}_{\text{ret}}\text{)}. \tag{12}$$

Using $\varepsilon = (x_t - \sqrt{\bar{\alpha}_t} x_0)/\sigma_t$, the conditional expectation of the training target given $x_t$ under this posterior becomes

$$\bar{\varepsilon}_t(x_t) := \mathbb{E}_{x_0^{(r)} \sim \pi_t(\cdot|x_t)} \left[ \frac{x_t - \sqrt{\bar{\alpha}_t} x_0^{(r)}}{\sigma_t} \right] = \sum_{x_0^{(r)} \in \mathcal{D}_{\text{ret}}} w_t(x_t; x_0^{(r)}) \cdot \frac{x_t - \sqrt{\bar{\alpha}_t} x_0^{(r)}}{\sigma_t}, \tag{13}$$

where $w_t(\cdot)$ is exactly the normalized weight in (12). Then the retain-only objective can be written as

$$L_{\text{ret}}(\theta) = \mathbb{E}_{t, x_t \sim p_{\text{ret},t}} \left[ \|\epsilon_{\theta}(x_t, t) - \bar{\varepsilon}_t(x_t)\|_2^2 \right] + \text{(variance term independent of } \theta\text{)}. \tag{14}$$

ReTrack's step is to change *how we sample* $x_t$ without changing the target objective: sample $x_t$ from $p_{\text{for},t}$ (thus focusing training near forget data), and rewrite the retain-only loss via importance sampling with the density ratio $p_{\text{ret},t}(x_t)/p_{\text{for},t}(x_t)$. This yields the unbiased reformulation used by ReTrack:

$$L_{\text{unlearn}}(\theta) := \mathbb{E}_{t, x_0^{(f)} \sim \mathcal{D}_{\text{for}}, x_t \sim q_t(\cdot|x_0^{(f)})} \left[ \frac{p_{\text{ret},t}(x_t)}{p_{\text{for},t}(x_t)} \sum_{x_0^{(r)} \in \mathcal{D}_{\text{ret}}} w_t(x_t; x_0^{(r)}) \left\| \epsilon_{\theta}(x_t, t) - \frac{x_t - \sqrt{\bar{\alpha}_t} x_0^{(r)}}{\sigma_t} \right\|_2^2 \right], \tag{15}$$

which is equivalent to $L_{\text{ret}}(\theta)$ up to the same $\theta$-independent variance term in (14). Note that the density ratio can be expressed as $p_{\text{ret},t}(x_t)/p_{\text{for},t}(x_t) = \sum_{x_0^{(r)}} q_t(x_t|x_0^{(r)})/(N_{\text{ret}} \cdot p_{\text{for},t}(x_t))$, and its product with $w_t$ yields unnormalized importance weights whose sum equals the density ratio $p_{\text{ret},t}(x_t)/p_{\text{for},t}(x_t)$. In other words, before approximation, this reformulation targets the same retain-only objective as LOGO retraining in the idealized limit.

**k-NN truncation from exponential weight decay (practical ReTrack).** Directly evaluating (13) is expensive because it sums over all retain samples. However, since $q_t(x_t|x_0)$ is Gaussian,

$$w_t(x_t; x_0^{(r)}) \propto q_t(x_t|x_0^{(r)}) \propto \exp\left(-\frac{\|x_t - \sqrt{\bar{\alpha}_t}x_0^{(r)}\|_2^2}{2\sigma_t^2}\right), \tag{16}$$

so weights decay exponentially with distance. Therefore $\bar{\varepsilon}_t(x_t)$ is well-approximated by retaining only the top-$K$ largest-weight terms, which correspond to the $K$ nearest neighbors under the Gaussian kernel induced by $q_t$ in $\mathcal{D}_{\mathrm{ret}}$. This produces ReTrack's interpretable redirection rule: "a noisy point generated from a forget example is trained to denoise toward nearby retain examples."

**Preservation / distillation term and deviation from the exact theory.** The unbiased equivalence to LOGO holds for the *full* objective (without truncation and with sufficient optimization). In practice, three approximations are introduced: (i) $K$-NN truncation in (13), (ii) finite-step fine-tuning, and (iii) an explicit preservation regularizer. ReTrack uses an interpolation with the vanilla retain-set loss as a regularizer; in our implementation we instead use the unified score-matching distillation loss (Eq. (7)), which acts as a stability regularizer that keeps $\theta_{-k}^{\mathrm{ul}}$ close to $\theta^{\mathrm{full}}$ on $\mathcal{D}_{\mathrm{ret}}$. This replacement improves stability and makes the preservation mechanism consistent across GUDA-U and GUDA-C, at the cost of introducing a controlled bias away from the exact LOGO optimum.

**Takeaway for GUDA-U.** Despite these practical approximations, ReTrack remains well-motivated for our purpose: among common unlearning objectives, it is uniquely derived as a (pre-approximation) unbiased reformulation of the retain-only training objective, hence directly aligned with the LOGO counterfactual target. Our paper uses ReTrack as a *computational proxy* for $\theta_{-k}^{\mathrm{logo}}$, and we validate the quality of this approximation empirically against LOGO in the main experiments.

## B. Method Details

### B.1. Conditional Unlearning Strategy for GUDA-C

For conditional text-to-image models, we implement the anchor construction $\mathcal{A}_{\mathrm{WSS}}$ via *weighted style selection*. The goal is to produce anchor conditions that remain compatible with the retain distribution after removing style $k$, while keeping the content of the forget sample unchanged.

**Descriptor extraction and CLIP embedding.** Given a forget prompt $c_f$ (from the style-descriptor template described in Appendix C.2.3), we parse (i) the content token (object) $o_f$ and (ii) a set of style descriptors $\mathcal{D}_f$. We compute a CLIP text embedding for the forget descriptors by averaging descriptor embeddings and L2-normalizing. We denote the resulting embedding by $e_f$.

**Style selection distribution.** We precompute a CLIP prototype embedding $e_s$ for each style $s$ using that style's descriptor set. We then define a distribution over retain styles $\mathcal{S}_{\mathrm{retain}}$:

$$\pi_s(s|c_f) = (1 - \eta_{\mathrm{mix}}) \frac{\exp(\tau e_f^\top e_s)}{\sum_{s' \in \mathcal{S}_{\mathrm{retain}}} \exp(\tau e_f^\top e_{s'})} + \eta_{\mathrm{mix}} \frac{1}{|\mathcal{S}_{\mathrm{retain}}|}, \tag{17}$$

where $\tau$ controls temperature sharpness and $\eta_{\mathrm{mix}}$ mixes uniform exploration. We set the probability of the forget style to zero by restricting to $\mathcal{S}_{\mathrm{retain}}$.

**Prompt synthesis.** We sample a retain style $s \sim \pi_s(\cdot|c_f)$ and synthesize an anchor prompt by combining the forget object $o_f$ with a small subset of descriptors from the sampled style:

$$c_a = \mathrm{Synthesize}(o_f, \mathcal{D}_s). \tag{18}$$

In our implementation, we randomly select a fixed number of descriptors (e.g., three) from $\mathcal{D}_s$ to form a style-descriptor prompt, and then encode it with the frozen text encoder to obtain the anchor text embedding. We use the forget latent as the anchor latent ($z_a = z_f$), so the anchor differs only in the conditioning signal while remaining on-distribution with respect to retain-style prompts.

**Example forget-to-anchor prompt pairs.** Table 15 provides concrete examples of the prompt substitutions produced by the anchor construction. These examples illustrate how style descriptors are replaced while preserving the object/content. The table is placed at the end of the Appendix for readability.

**Optimization.** We use the common preservation loss (Eq. (7)) with stabilization weight $\lambda_{\text{pres}} = 2.0$, learning rate $2 \times 10^{-6}$, and optimize for 2,000 fine-tuning steps per style. This yields an unlearned model that removes the target style's influence while maintaining retain-style behavior. We set $\tau = 2.0$ and $\eta_{\text{mix}} = 0.1$ based on a hyperparameter sweep over temperature and mixing weight.

**Ablation: anchor strategy and style-sampling mode.** To validate the two key design choices of $\mathcal{A}_{\text{WSS}}$, we compare three anchor variants under identical hyperparameters (lr $= 2 \times 10^{-6}$, $\lambda_{\text{pres}} = 2.0$, $\eta_{\text{mix}} = 0.1$, 2,000 steps). Table 3 shows that AWSS consistently outperforms both simpler alternatives across all six metrics. Replacing CLIP-weighted style selection with uniform random selection over retain styles reduces performance on all metrics, confirming that semantic proximity of the anchor matters. Replacing the retain-style anchor with a style-removed condition (which strips style descriptors from the forget prompt rather than substituting retain-style descriptors) degrades all metrics more substantially, indicating that redirecting toward a semantically compatible retain-style direction is more effective than simply removing style information. These results support the view that AWSS provides a more faithful approximation to the conditional LOGO target than simpler anchor alternatives.

*Table 3.* Anchor strategy ablation on UnlearnCanvas (16 styles, 320 queries). All rows use identical hyperparameters (lr $= 2 \times 10^{-6}$, $\lambda_{\text{pres}} = 2.0$, step 2000). [†]Same configuration as Table 2. Best in **bold**; ↑ higher is better.

| Anchor variant | Top-1↑ | MRR↑ | NDCG@3↑ | Top-3↑ | RBO↑ | Spearman↑ |
|---|---|---|---|---|---|---|
| AWSS (proposed)[†] | **0.456** | **0.582** | **0.734** | **0.405** | **0.446** | **0.239** |
| Uniform sampling | 0.447 | 0.552 | 0.680 | 0.332 | 0.410 | 0.104 |
| Style removed | 0.400 | 0.526 | 0.671 | 0.331 | 0.409 | 0.118 |

# C. Experimental Setup

**Remark on loss weighting conventions.** Throughout this paper, we use two different conventions for balancing forget and preservation losses, reflecting the implementations used in our experiments:

- **UnlearnCanvas (conditional setting)**: We use $\mathcal{L}_{\text{unlearn}} = \mathcal{L}_{\text{forget}} + \lambda_{\text{pres}}\mathcal{L}_{\text{preserve}}$, where the preservation loss is weighted by $\lambda_{\text{pres}} = 2.0$. This follows the convention in Eq. (6) and is used in all UnlearnCanvas experiments (Sec. 4.2.2, Appendix B.1).

- **CIFAR-10 (unconditional setting)**: We use $\mathcal{L}_{\text{unlearn}} = \lambda_{\text{forget}}\mathcal{L}_{\text{forget}} + \mathcal{L}_{\text{preserve}}$, where the forget loss is weighted by $\lambda_{\text{forget}} = 0.03$ for ReTrack (or $\lambda_{\text{forget}} = 0.3$ for ESD), and the preservation loss has unit weight. This is equivalent to the first convention by reparameterization, but reflects the actual implementation used for CIFAR-10.

Both conventions are mathematically equivalent (related by reciprocal transformation), but we report the hyperparameters as they were used in the respective implementations to ensure reproducibility.

## C.1. CIFAR-10 Experiments

### C.1.1. DATASET AND PREPROCESSING

Unconditional training uses the CIFAR-10 training split (50,000 images) for training and the test split (10,000 images) for validation. Unlearning uses train split for retain/forget subsets and test split for validation.

### C.1.2. MODEL ARCHITECTURE & TRAINING CONFIGURATION

Table 4 summarizes our model architecture and training configuration. We implement our diffusion model using the Diffusers library (von Platen et al., 2022).

**Exposure-matched training.** In the unconditional setting, the all-group model excludes one random class per epoch to match the expected training set size of LOGO models, while LOGO models exclude a fixed target class throughout training. This ensures fair comparison by equalizing the number of weight updates across models. Evaluation uses the epoch-2,400 checkpoints for all models ($\theta^{\text{full}}$ and all $\theta^{\text{logo}}_{-k}$).

*Table 4.* Model architecture and training configuration.

| Parameter | Value |
|---|---|
| *UNet Architecture* | |
| Sample size | 32 |
| In / Out channels | 3 / 3 |
| Layers per block | 3 |
| Block out channels | (128, 256, 256, 256) |
| Dropout | 0.3 |
| Attention resolutions | 16, 8 |
| *Diffusion Process* | |
| Scheduler | DDPM |
| Timesteps | 4,000 |
| Noise schedule | Squared cosine |
| Prediction type | $\epsilon$-prediction |
| Loss function | MSE |
| *Training* | |
| Optimizer | AdamW |
| Weight decay | $10^{-4}$ |
| Learning rate | $10^{-4}$ |
| LR schedule | Cosine (5,000 warmup) |
| Gradient clipping | 1.0 |
| Batch size | 256 |
| Epochs | 2,400 |

### C.1.3. UNLEARNING IMPLEMENTATION

**ReTrack unlearning**: Uses paired batches (one retain, one forget) per optimization step. Forget loss redirects denoising targets toward $k$-nearest neighbors ($k = 10$) from the retain set, weighted by importance sampling. Distance is computed in pixel space using the Euclidean metric. KL trust-region clipping (`kl_cap=1.0`) stabilizes training. Retention loss uses teacher-student distillation to preserve retain-set behavior. Hyperparameters: $\lambda_{\text{forget}} = 0.03$, `learning_rate`$= 1 \times 10^{-5}$, epochs $= 20$, timestep range restricted to $[2000, 3900]$.

**ESD unlearning**: Forget loss uses negative guidance with `guidance_weight`$= 5.0$, requiring both conditional and unconditional models. Retention loss uses distillation with the unconditional teacher model. Hyperparameters: $\lambda_{\text{forget}} = 0.3$, `learning_rate`$= 3 \times 10^{-5}$, epochs $= 20$.

**Unlearned models**: 10 ReTrack models and 10 ESD models created by unlearning each class from the all-group model, using the common score-matching preservation loss (Eq. (7)) in both cases.

### C.1.4. GRADIENT-BASED METHOD CONFIGURATION

We reimplemented D-TRAK (Zheng et al., 2024) and TRAK (Park et al., 2023) based on the official D-TRAK repository[2], adhering to the hyperparameters specified in the original papers.

**Feature extraction**: Gradients computed for all trainable UNet parameters with respect to per-timestep loss. D-TRAK uses mean-squared L2 norm of predicted noise; TRAK uses MSE between predicted and ground-truth noise.

**Timestep sampling**: Fixed uniform grid of 10 timesteps over $[0, T)$ with deterministic per-timestep noise seeding for reproducibility.

**Random projection**: Gaussian random projection to 32,768 dimensions with fixed seed.

**Ridge regularization**: D-TRAK baseline $\lambda = 1000$; TRAK baseline $\lambda = 100$. We evaluate $\lambda$ sweep at $0.1\times$, $1\times$, and $10\times$ baseline values.

**Group aggregation**: We implement multiple aggregation strategies (sum, mean, max, top-k mean) to convert instance-level scores to group-level attribution. The main paper reports sum aggregation; ablations on aggregation strategies are provided

---

[2]https://github.com/sail-sg/D-TRAK

in Appendix D.2.

### C.1.5. DAS CONFIGURATION

We implement Diffusion Attribution Score (DAS) (Lin et al., 2025) following the DAS1 variant with residual correction and diagonal adjustment.

**Feature extraction**: Gradients computed for all trainable UNet parameters with respect to per-timestep mean loss. We use 10 uniformly sampled timesteps for feature extraction and project to 32,768 dimensions using Gaussian random projection with fixed seed.

**Error (residual) computation**: We compute residuals (prediction errors) using 1,000 uniformly sampled timesteps to capture detailed attribution signals across the diffusion process.

**Ridge regularization**: We evaluate $\lambda \in \{0.1, 0.01, 0.001\}$ and report $\lambda = 0.01$ in the main paper, which provides the best balance between regularization strength and attribution accuracy.

**Group aggregation**: We implement multiple aggregation strategies (sum, mean, max, median, top-1 mean, top-10 mean) to convert instance-level scores to group-level attribution. The main paper reports mean aggregation ($\lambda = 0.01$); ablations on both $\lambda$ and aggregation strategies are provided in Appendix D.2.

### C.1.6. EVALUATION PROTOCOL

**Generated samples**: 2,048 images sampled from the all-group model using DDPM sampler with 4,000 inference steps.

**ELBO estimation**: For each generated image, we compute ELBO via the sum of per-timestep KL divergence between the true posterior $q(x_{t-1}|x_t, x_0)$ and model posterior $p_\theta(x_{t-1}|x_t)$. Timesteps are sampled on a uniform grid with stride 10. All methods (LOGOA, ReTrack, ESD) are evaluated using the same timestep range $[1, 4000]$ to ensure fair comparison. Variance reduction is achieved by reusing the same per-timestep noise across base and counterfactual models.

**CLIPA baseline**: Uses CLIP (openai/clip-vit-base-patch32)[3] to compute mean embeddings for 100 training images per class. Attribution score is the cosine similarity between generated image embedding and class prototype embedding.

**Evaluation metrics**: For each generated image, we rank all 10 classes by their attribution scores and compute correlation metrics against the LOGOA gold-standard ranking. We report NDCG@3, MRR, RBO (with $p = 0.9$), Top-1 agreement, and Top-3 set overlap, and Spearman correlation across all 2,048 samples.

**Qualitative generations under group removal.** Figure 3 shows generations from the all-class model paired with LOGO, ReTrack, and ESD counterfactual models. We use DDIM sampling with 100 steps and identical initial noise, and show removals of class 1 (automobile) and class 7 (horse).

### C.2. UnlearnCanvas Experiments

#### C.2.1. DATASET AND STYLES

The UnlearnCanvas dataset (Zhang et al., 2024b) contains approximately 24,400 images spanning 60 artistic styles plus 400 style-free seed images. Each of the 60 styles includes 400 images (20 objects $\times$ 20 images per object), and an additional 400 seed images (20 objects $\times$ 20 images) without any artistic style are provided. This cross-style correspondence with shared object categories enables fine-grained style attribution analysis. For computational feasibility, we evaluate attribution on the first 16 styles in alphabetical order. The all-group model $\theta^{\text{full}}$ is obtained by fine-tuning Stable Diffusion 1.5 (Rombach et al., 2022) on all 60 styles, following the UnlearnCanvas protocol (Zhang et al., 2024b). Each LOGO model $\theta^{\text{logo}}_{-k}$ ($k \in \{1, \ldots, 16\}$) is similarly fine-tuned from Stable Diffusion 1.5 on 59 styles (removing only target style $k$), and each unlearned model $\theta^{\text{ul}}_{-k}$ is fine-tuned from $\theta^{\text{full}}$ to remove style $k$. Thus, the counterfactual models correspond to a 60-style world with one style removed; the 16-style restriction applies only to evaluation (which styles we compute attribution scores for), not to model training.

The 16 selected styles are: Abstractionism, Artist Sketch, Blossom Season, Blue Blooming, Bricks, Byzantine, Cartoon, Cold Warm, Color Fantasy, Comic Etch, Crayon, Crypto Punks, Cubism, Dadaism, Dapple, and Defoliation. These styles

---

[3]https://huggingface.co/openai/clip-vit-base-patch32

**CIFAR-10 All-class & LOGO models**

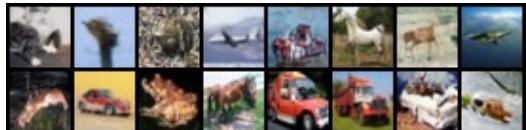

all-class model: $\theta^{\mathrm{full}}$

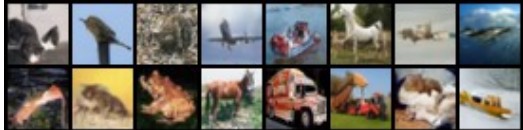

LOGO model (class 1: automobile) : $\theta^{\mathrm{logo}}_{-(k=1)}$

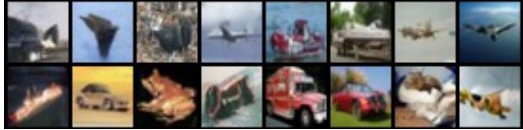

LOGO model (class 7: horse) : $\theta^{\mathrm{logo}}_{-(k=7)}$

*(a)* All-class model and LOGO models.

**CIFAR-10 All-class & Unlearned models (ReTrack)**

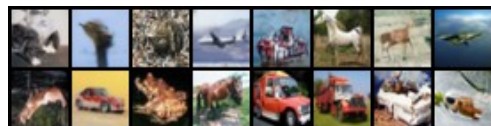

all-class model: $\theta^{\mathrm{full}}$

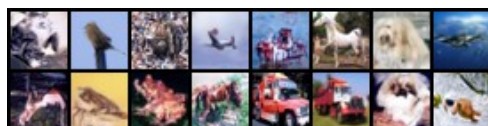

Unlearned model (ReTrack, class 1: automobile) : $\theta^{\mathrm{ul}}_{-(k=1)}$

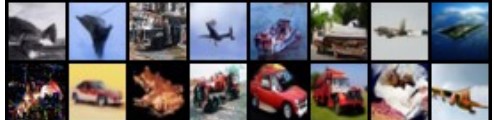

Unlearned model (ReTrack, class 7: horse) : $\theta^{\mathrm{ul}}_{-(k=7)}$

*(b)* All-class model and ReTrack models.

**CIFAR-10 All-class & Unlearned models (ESD)**

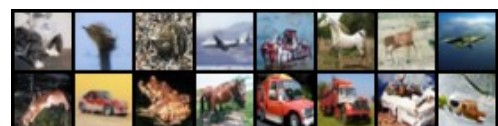

all-class model: $\theta^{\mathrm{full}}$

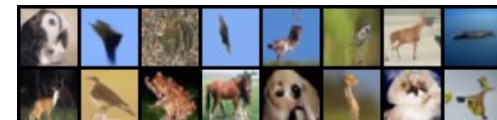

Unlearned model (ESD, class 1: automobile) : $\theta^{\mathrm{ul}}_{-(k=1)}$

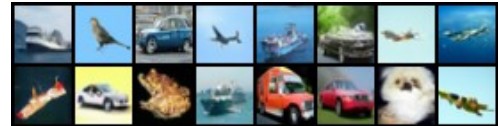

Unlearned model (ESD, class 7: horse) : $\theta^{\mathrm{ul}}_{-(k=7)}$

*(c)* All-class model and ESD models.

*Figure 3.* CIFAR-10 generations under group removal. Each panel pairs the all-class model with a counterfactual model trained without class 1 (automobile) or class 7 (horse). We use DDIM with 100 steps and the same initial noise for each pair. The top row shows LOGO models, and the bottom row shows ReTrack and ESD unlearned models.

are selected alphabetically from the full 60-style set and span diverse visual characteristics.

**Dataset structure**: Images are organized as `{style}/{object}/{idx}.jpg` with 20 object categories (Architectures, Bears, Birds, Butterfly, Cats, Dogs, Fishes, Flame, Flowers, Frogs, Horses, Human, Jellyfish, Rabbits, Sandwiches, Sea, Statues, Towers, Trees, Waterfalls). The dataset is available on HuggingFace[4].

**Preprocessing**: All training images are resized to $512 \times 512$ with center cropping. Horizontal flips are applied during training for augmentation. Pixel values are normalized to $[-1, 1]$ range. No explicit train/validation split; all images used for training.

---

[4]https://huggingface.co/datasets/OPTML-Group/UnlearnCanvas

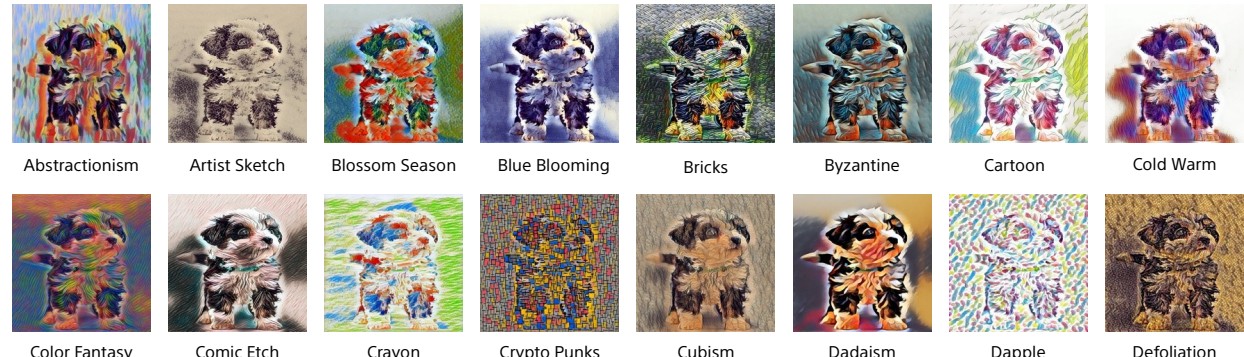

*Figure 4.* UnlearnCanvas training data examples for the Dogs class across the 16 evaluation styles. Images within the same style exhibit consistent visual characteristics, supporting style-level attribution analysis.

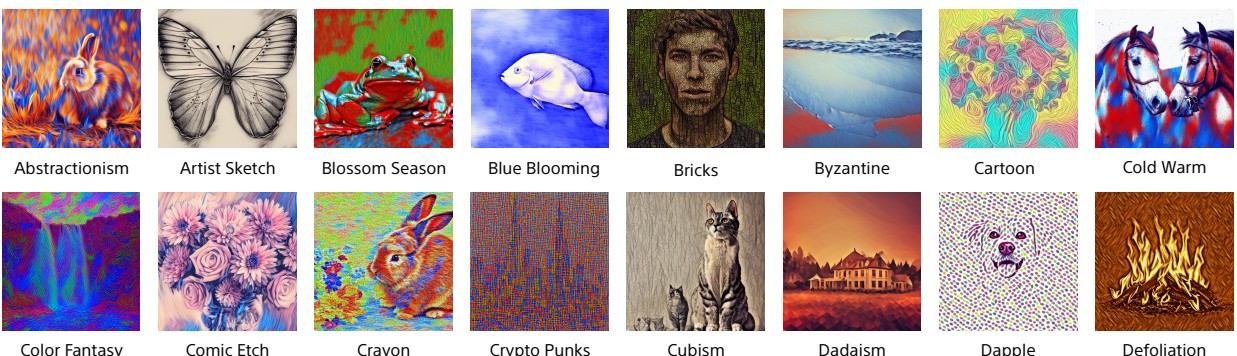

*Figure 5.* Evaluation images generated by the all-group model using the evaluation prompts. Due to partial descriptor overlap across styles, some generations reflect a dominant style while others exhibit mixed visual characteristics.

### C.2.2. TRAINING CONFIGURATION

**Base model**: Stable Diffusion 1.5 (Rombach et al., 2022) (`runwayml/stable-diffusion-v1-5`) with frozen text encoder and VAE, full fine-tuning of UNet. Training uses DDPMScheduler with default configuration and $\epsilon$-prediction type. Loss function is mean squared error between predicted and ground-truth noise.

**All-group model**: Trained on all 60 styles for 10,000 steps with learning rate $1 \times 10^{-6}$ (constant schedule, no warmup), batch size 32, gradient accumulation steps 1, mixed precision training (fp16), AdamW optimizer ($\beta_1 = 0.9$, $\beta_2 = 0.999$, weight decay 0.01, $\epsilon = 10^{-8}$), gradient clipping with max norm 1.0. EMA enabled for training stability.

**LOGO models**: 16 models, each trained on 59 styles (excluding one target style) for 10,000 steps from the base SD 1.5 checkpoint using identical hyperparameters to ensure exposure-matched training conditions. This ensures that each LOGO model receives the same total number of training steps as the all-group model, avoiding confounding from different optimization budgets. No explicit style balancing; dataloader uses standard shuffling over all retain samples.

**Remark on the LOGO oracle (implementation).** While Sec. 3.3 defines $\theta_{-k}^{\text{logo}}$ as the counterfactual trained on $\mathcal{D}_{-k}$, in UnlearnCanvas we implement this oracle by fine-tuning from the same SD 1.5 initialization as the full model under an exposure-matched recipe and budget. This yields a practical LOGO proxy appropriate for large pretrained diffusion models, and is the oracle used in our evaluation.

**Unlearned models**: 16 models created via the conditional unlearning procedure (GUDA-C, Sec. 4.2.2) from the all-group model, using weighted style-selection anchors for redirection. Fine-tuning uses learning rate $2 \times 10^{-6}$, batch size 4, and 2,000 optimization steps per style. Forget loss redirects style-specific conditioning toward CLIP-weighted style-selection anchors. Preservation loss uses stabilization weight $\lambda_{\text{pres}} = 2.0$ (Eq. (7)). CLIP prototype embeddings are precomputed

using `openai/clip-vit-large-patch14`[5] with style selection parameters $\tau = 2.0$ (temperature sharpness) and $\eta_{\mathrm{mix}} = 0.1$ (uniform mixing weight for exploration).

### C.2.3. TRAINING AND EVALUATION PROMPT DESIGN

To evaluate generalization beyond the training distribution, we intentionally introduce distribution shift between training and evaluation prompts. This tests whether attribution methods capture the true group-removal counterfactual influence of style training data rather than superficial prompt pattern matching.

**Motivation for style-descriptor prompts.** UnlearnCanvas images were originally captioned with simple templates like "{object} in {style} style", which conflate style identity (the style name) with visual characteristics. Such prompts are insufficient for attribution evaluation because: (i) they provide minimal stylistic guidance beyond the label, making it difficult to distinguish visual properties from label association, and (ii) they prevent testing whether models learn causal visual features versus memorizing label-content correlations. We therefore construct prompts that describe visual style properties (e.g., brushwork, color palette, lighting) using style-specific descriptors extracted from training images via a vision-language model (Qwen2.5-VL (Bai et al., 2025)).

**Descriptor extraction and filtering.** For each style, we sample training images and prompt the vision-language model to generate visual style descriptors while excluding content/object information. Raw descriptors are normalized to canonical forms and filtered via a soft-ranking criterion: $\mathrm{score}(d, s) = \frac{\mathrm{freq}(d,s)}{\mathrm{support}(d)^{\alpha}}$, where $\mathrm{freq}(d, s)$ is descriptor $d$'s occurrence count within style $s$, $\mathrm{support}(d)$ is the number of styles where $d$ appears at least a minimum frequency threshold, and $\alpha \in (0, 1]$ controls the penalty for cross-style sharing. We use $\alpha = 0.5$ with a maximum support cap of 20 styles, allowing shared atmospheric terms (e.g., "watercolor", "impasto") while preserving style distinctiveness. The top-5 scored descriptors per style form the final descriptor vocabulary.

**Training prompts.** We construct prompts by combining a randomized subject template (e.g., "A detailed view of {object}", "A depiction of {object}") with 3 sampled descriptors from the style-specific vocabulary: "{subject}, artistic style featuring {descriptor1}, {descriptor2}, {descriptor3}". This preserves visual style information while diversifying surface structure to prevent template overfitting.

**Evaluation prompts.** Evaluation prompts use the same descriptor vocabulary but apply: (i) a different template ("An artwork of {object}, with {shuffled descriptors}"), (ii) complete omission of style names, and (iii) deterministic descriptor-order shuffling via a hash of (style, object) to prevent training-sequence memorization. This ensures attribution methods must rely on understanding the influence of training data on model behavior rather than exploiting prompt patterns.

Figure 4 shows example UnlearnCanvas training images for a fixed object class (Dogs) across the 16 evaluation styles, illustrating that images within a style share consistent visual characteristics. Figure 5 shows evaluation images generated by the all-group model using the above evaluation prompts, where descriptor overlap across styles can yield both strongly style-specific generations and images that visually mix multiple styles. Prompt examples are summarized in Table 14 at the end of the Appendix.

### C.2.4. EVALUATION IMAGE GENERATION

We generate 320 evaluation images (20 per style for 16 styles) from the all-group model to assess attribution quality. Generation uses DPMSolverMultistepScheduler with 50 inference steps, guidance scale 7.5, and seed 42 (fixed generator initialization for each image). No negative prompts were used. Output images saved at default resolution without explicit height/width specification.

### C.2.5. ELBO ESTIMATION PROTOCOL

For each generated image, we compute ELBO via per-timestep KL divergence $\sum_t \mathrm{KL}(q(x_{t-1}|x_t, x_0) \| p_\theta(x_{t-1}|x_t))$ between the true posterior (using the generated image as $x_0$) and the model's learned reverse process.

**Timestep sampling**: Uniform grid with stride 10 over $t \in [10, 999]$, yielding 100 timesteps per ELBO computation.

**Noise sampling**: Single noise sample per timestep, with the same noise reused across all models (all-group, LOGO, unlearned) for variance reduction.

---

[5] https://huggingface.co/openai/clip-vit-large-patch14

**Precision**: UNet forward pass uses mixed precision (bf16 via autocast), while VAE and text encoder use fp16. KL divergence computed in fp32 for numerical stability. Note that ELBO computation is latent-space only; the VAE likelihood term is not included.

### C.2.6. BASELINE IMPLEMENTATION DETAILS

**LOGOA (gold standard)**: ELBO difference between all-group model and each LOGO model (Eq. (4)), computed over the timestep grid described above. Positive values indicate that excluding the style reduces the model's ability to explain the generated image, quantifying the style's group-removal counterfactual contribution.

**GUDA**: ELBO difference between all-group model and each unlearned model (Eq. (5)), using identical timestep sampling, noise seeding, and precision settings as LOGOA for fair comparison.

**CLIPA**: Semantic similarity baseline using CLIP (`openai/clip-vit-large-patch14`) (Radford et al., 2021)[6]. For each style, we precompute mean embeddings from 400 training images. Attribution score is the cosine similarity between the generated image embedding and the style prototype (mean) embedding. This measures semantic association in embedding space rather than counterfactual influence through data removal.

**Wang et al. (2024) baseline**: Instance-level unlearning approach adapted for group attribution, reimplemented based on the official repository[7]. We precompute training latents for all images with horizontal flip augmentation (yielding $2\times$ samples per image), using resizing to 512 and center cropping. Empirical Fisher information is computed for cross-attention keys and values over 5 epochs with batch size 16 in fp32 precision. For each generated image, we sample 20 base samples per style (320 total) with flipped versions (640 total), using per-sample noise seeding based on global index. Unlearning hyperparameters: learning rate 0.1, 100 steps, 10 time samples per forward pass. Attribution is computed at checkpoints [10, 25, 50, 100] steps. Instance-level influences are aggregated to style-level by summing within each style group (flip variants combined via max operation).

## C.3. Evaluation Metrics

We evaluate attribution quality using multiple metrics that capture different aspects of ranking agreement with the LOGOA gold standard. These metrics are selected to emphasize correctness at the top of the ranking, as identifying the most influential groups matters most for practical applications (copyright assessment, fair compensation, model debugging).

### C.3.1. TOP-1 AGREEMENT

Fraction of images where the method's top-ranked style exactly matches the gold-standard top-1 style:

$$\text{Top-1} = \mathbb{E}\left[\mathbb{1}[\text{method rank}_1 = \text{gold rank}_1]\right]. \tag{19}$$

Range: $[0, 1]$, higher is better. This is the most direct measure of whether the method identifies the single most influential group.

### C.3.2. MEAN RECIPROCAL RANK (MRR)

MRR quantifies "how far the method missed the gold-standard Top-1". For each image, we compute the reciprocal rank of the gold top-1 style in the predicted ranking:

$$\text{RR}(x_0) = \frac{1}{\text{rank}_s(c^*)}, \tag{20}$$

where $c^*$ is the gold-standard Top-1 style and $\text{rank}_s(c^*)$ is its rank in the predicted ranking (1-based). MRR is the average reciprocal rank across all images. Range: $(0, 1]$, higher is better. MRR $= 1.0$ indicates perfect Top-1 identification; MRR $= 0.5$ means the gold-standard Top-1 is typically ranked 2nd.

---

[6] `https://huggingface.co/openai/clip-vit-large-patch14`
[7] `https://github.com/peterwang512/AttributeByUnlearning`

### C.3.3. NDCG@3 (NORMALIZED DISCOUNTED CUMULATIVE GAIN)

NDCG@3 (Järvelin & Kekäläinen, 2002) is a top-heavy ranking quality metric that emphasizes correctness at top positions with position-dependent discounting. We first compute Discounted Cumulative Gain at depth $k = 3$:

$$\text{DCG@3} = \sum_{r=1}^{3} \frac{2^{\text{rel}[\pi_s(r)]} - 1}{\log_2(r+1)}, \tag{21}$$

where $\pi_s$ is the permutation induced by predicted scores (descending order) and $\text{rel}$ is the relevance vector derived from gold scores (we use rank-based relevance: $\text{rel}_i = N - \text{rank}_g(i) + 1$ for robustness, where $N$ is the total number of styles). NDCG@3 normalizes by the ideal DCG:

$$\text{NDCG@3} = \frac{\text{DCG@3}}{\text{IDCG@3}}, \tag{22}$$

where IDCG@3 is computed using the gold ranking. Range: $[0, 1]$, higher is better. NDCG@3 $\approx 0.9$ indicates strong agreement on the top-3 most influential styles.

### C.3.4. TOP-3 OVERLAP

Average overlap between the method's top-3 styles and the gold-standard top-3 styles, measured as set intersection divided by 3:

$$\text{Top-3} = \mathbb{E}\left[\frac{|\{\text{method top-3}\} \cap \{\text{gold top-3}\}|}{3}\right]. \tag{23}$$

Range: $[0, 1]$, higher is better. This metric does not require order matching within the top-3 sets. Note that Top-3 overlap can be lower than Top-1 agreement when the method correctly identifies the single most influential style but differs from the gold standard on positions 2–3.

### C.3.5. RBO (RANK-BIASED OVERLAP)

RBO (Webber et al., 2010) is a top-weighted ranking similarity metric with geometric decay. For finite lists of length $N$ (total number of styles), we use the truncated form:

$$\text{RBO} = (1 - p) \sum_{d=1}^{N} p^{d-1} X_d, \tag{24}$$

where $X_d = |A_d \cap B_d|/d$ is the overlap fraction at depth $d$, $A_d$ and $B_d$ are the top-$d$ items in predicted and gold rankings, and $p \in (0, 1)$ is a decay parameter. We use $p = 0.9$, which emphasizes top ranks while accounting for deeper positions. Range: $[0, 1 - p^N]$ (maximum $1 - p^N \approx 0.815$ for $N = 16$), higher is better. RBO complements NDCG@3 with a different weighting scheme for top-heavy evaluation.

### C.3.6. SPEARMAN CORRELATION

Spearman's rank correlation coefficient measures monotonic relationship between predicted and gold-standard rankings across all $N$ styles ($N = 10$ for CIFAR-10, $N = 16$ for UnlearnCanvas). Unlike the metrics above, Spearman weights all ranking positions equally, providing a complementary view of global ranking agreement. Range: $[-1, 1]$, higher is better.

## D. Additional Results

### D.1. ReTrack Ablation Studies

We report ablations for GUDA-U on CIFAR-10 using the LOGOA evaluation protocol. Unless noted, results use the same attribution computation as in the main experiment. The main configuration ($\text{lr} = 1 \times 10^{-5}$, epochs $= 20$, $\lambda = 0.03$, $K = 10$) is highlighted in gray in each table.

*Table 5.* ReTrack ablation: effect of unlearning epochs (lr $= 1 \times 10^{-5}$, $\lambda = 0.03$, $K = 10$). Best in **bold**; $\uparrow$ higher is better. The main configuration is shaded.

| Epochs | Top-1↑ | MRR↑ | NDCG@3↑ | Top-3↑ | RBO↑ | Spearman↑ |
|---|---|---|---|---|---|---|
| 10 | 0.610 | 0.727 | 0.632 | 0.459 | 0.408 | 0.254 |
| 20 | 0.727 | **0.798** | **0.677** | **0.475** | **0.423** | **0.265** |
| 30 | **0.728** | 0.794 | 0.670 | 0.471 | 0.421 | 0.258 |
| 40 | 0.723 | 0.790 | 0.670 | 0.474 | 0.421 | 0.262 |
| 50 | 0.717 | 0.785 | 0.664 | 0.468 | 0.420 | 0.257 |

*Table 6.* ReTrack ablation: effect of forget loss weight $\lambda$ (lr $= 1 \times 10^{-5}$, epochs $= 20$, $K = 10$). Best in **bold**; $\uparrow$ higher is better. The main configuration is shaded.

| $\lambda$ | Top-1↑ | MRR↑ | NDCG@3↑ | Top-3↑ | RBO↑ | Spearman↑ |
|---|---|---|---|---|---|---|
| 0.003 | 0.520 | 0.636 | 0.553 | 0.423 | 0.384 | 0.191 |
| 0.01 | 0.702 | 0.774 | 0.656 | 0.465 | 0.417 | 0.251 |
| 0.03 | **0.727** | **0.798** | **0.677** | **0.475** | **0.423** | **0.265** |

*Table 7.* ReTrack ablation: effect of $K$-nearest neighbors (lr $= 1 \times 10^{-5}$, $\lambda = 0.03$, epochs $= 20$). Best in **bold**; $\uparrow$ higher is better. The main configuration is shaded.

| $K$ | Top-1↑ | MRR↑ | NDCG@3↑ | Top-3↑ | RBO↑ | Spearman↑ |
|---|---|---|---|---|---|---|
| 1 | 0.725 | 0.797 | **0.677** | 0.476 | **0.423** | **0.269** |
| 5 | 0.726 | 0.797 | **0.677** | 0.478 | **0.423** | 0.268 |
| 10 | **0.727** | **0.798** | **0.677** | 0.475 | **0.423** | 0.265 |
| 20 | 0.720 | 0.794 | 0.673 | 0.473 | 0.422 | 0.264 |
| 50 | **0.727** | **0.798** | 0.676 | 0.474 | **0.423** | 0.266 |

### D.1.1. EFFECT OF UNLEARNING EPOCHS

Table 5 varies the number of unlearning epochs while fixing lr $= 1 \times 10^{-5}$ and $\lambda = 0.03$ with $K = 10$. Performance improves sharply from 10 to 20 epochs, then saturates. The 20-epoch setting yields the best NDCG@3 and competitive head metrics, indicating that modest unlearning budgets are sufficient for strong counterfactual approximation.

### D.1.2. EFFECT OF FORGET LOSS WEIGHT $\lambda$

Table 6 varies $\lambda$ with lr $= 1 \times 10^{-5}$ and epochs $= 20$. Increasing $\lambda$ strengthens forgetting and improves head metrics up to $\lambda = 0.03$. This suggests that stronger redirection is beneficial under the current preservation loss, without degrading ranking stability.

### D.1.3. EFFECT OF $K$-NEAREST NEIGHBORS

Table 7 varies the number of nearest neighbors $K$ used in the importance-weighted target approximation (Eq. (8)), with lr $= 1 \times 10^{-5}$, $\lambda = 0.03$, and epochs $= 20$. ReTrack truncates the full importance-weighted sum over $\mathcal{D}_{-k}$ to the $K$ nearest neighbors for computational efficiency. The results show that performance is relatively stable across $K \in \{1, 5, 10, 20, 50\}$, with $K = 10$ achieving the best NDCG@3 and competitive head metrics. This indicates that a modest number of neighbors suffices to approximate the full posterior target, validating the truncated $K$-NN strategy.

### D.1.4. EFFECT OF LEARNING RATE

Table 8 varies the learning rate with $\lambda = 0.03$ and epochs $= 20$. The lr $= 1 \times 10^{-5}$ setting provides the best overall balance across head metrics, while larger rates yield similar Top-1 but slightly weaker NDCG@3 and MRR. This indicates that conservative step sizes stabilize the redirection objective under the shared preservation loss.

*Table 8.* ReTrack ablation: effect of learning rate (epochs $= 20$, $\lambda = 0.03$, $K = 10$). Best in **bold**; $\uparrow$ higher is better. The main configuration is shaded.

| Learning rate | Top-1$\uparrow$ | MRR$\uparrow$ | NDCG@3$\uparrow$ | Top-3$\uparrow$ | RBO$\uparrow$ | Spearman$\uparrow$ |
|---|---|---|---|---|---|---|
| $3 \times 10^{-4}$ | 0.699 | 0.783 | 0.665 | 0.463 | 0.419 | 0.264 |
| $1 \times 10^{-4}$ | **0.729** | 0.797 | 0.673 | 0.470 | 0.422 | 0.257 |
| $3 \times 10^{-5}$ | 0.721 | 0.790 | 0.671 | 0.473 | 0.420 | 0.253 |
| $1 \times 10^{-5}$ | 0.727 | **0.798** | **0.677** | **0.475** | **0.423** | **0.265** |

*Table 9.* DAS ablation: effect of ridge regularization $\lambda$ (mean aggregation fixed). Best in **bold**; $\uparrow$ higher is better.

| $\lambda$ | Top-1$\uparrow$ | MRR$\uparrow$ | NDCG@3$\uparrow$ | Top-3$\uparrow$ | RBO$\uparrow$ | Spearman$\uparrow$ |
|---|---|---|---|---|---|---|
| 0.1 | 0.293 | 0.514 | 0.489 | 0.410 | 0.360 | 0.187 |
| 0.01 | **0.716** | **0.794** | **0.675** | **0.473** | **0.422** | **0.267** |
| 0.001 | 0.715 | 0.794 | 0.674 | 0.471 | 0.422 | 0.267 |

*Table 10.* DAS ablation: effect of aggregation strategies ($\lambda = 0.01$ fixed). Best in **bold**; $\uparrow$ higher is better.

| Aggregation | Top-1$\uparrow$ | MRR$\uparrow$ | NDCG@3$\uparrow$ | Top-3$\uparrow$ | RBO$\uparrow$ | Spearman$\uparrow$ |
|---|---|---|---|---|---|---|
| mean | **0.716** | **0.794** | **0.675** | 0.473 | **0.422** | 0.267 |
| max | 0.625 | 0.742 | 0.647 | 0.470 | 0.412 | 0.260 |
| median | 0.655 | 0.744 | 0.631 | 0.443 | 0.406 | 0.210 |
| top-10 | 0.693 | 0.784 | 0.674 | **0.480** | **0.422** | **0.278** |

## D.2. DAS Ablation Studies

We conduct ablation studies on the Diffusion Attribution Score (DAS) (Lin et al., 2025) method to investigate the impact of (i) ridge regularization parameter $\lambda$ and (ii) aggregation strategies for converting instance-level scores to group-level attribution.

### D.2.1. EFFECT OF RIDGE REGULARIZATION PARAMETER $\lambda$

Table 9 presents the effect of ridge regularization parameter $\lambda \in \{0.1, 0.01, 0.001\}$ with mean aggregation fixed. The results show that $\lambda = 0.01$ (highlighted) achieves the best performance across all metrics, with Top-1 agreement of 71.6% and NDCG@3 of 0.675. Using $\lambda = 0.1$ leads to over-regularization, substantially degrading performance (Top-1 drops to 29.3%). Conversely, $\lambda = 0.001$ provides slightly weaker regularization but maintains comparable performance to $\lambda = 0.01$, suggesting that the method is relatively robust within the range $[0.001, 0.01]$. This validates our choice of $\lambda = 0.01$ for the main experiments.

### D.2.2. EFFECT OF AGGREGATION STRATEGIES

Table 10 presents the effect of different aggregation strategies for converting instance-level DAS scores to group-level attribution, with $\lambda = 0.01$ fixed. We compare four strategies: mean (averaging all instance scores in a group), max (taking the maximum score), median (taking the median score), and top-10 (averaging the top-10 instance scores).

The results reveal that mean and top-10 aggregation strategies perform best. This suggests that averaging over multiple instances provides more robust group-level signals than selecting individual instances (top-1) or using order statistics (max, median). The max strategy achieves 62.5% Top-1 agreement, indicating that the single most influential instance per group provides some signal but is less reliable than averaging. The median strategy (65.5% Top-1) falls between max and mean, suggesting that central tendency measures benefit from considering multiple instances but are less effective than mean aggregation.

*Table 11.* Total computational cost for CIFAR-10 attribution (10 classes, 2,048 query images). All methods executed on single NVIDIA H100 80GB GPU.

| Method | Preprocessing | Query-time | Total |
|---|---|---|---|
| GUDA (Ours) | 1:07 | 0:55 | 2:02 |
| DAS | 28:47 | 6:36 | 35:24 |
| LOGOA (oracle) | 206:52 | 0:55 | 207:47 |

*Table 12.* Total computational cost for UnlearnCanvas attribution (16 styles, 320 query images). All methods executed on single NVIDIA H100 80GB GPU.

| Method | Preprocessing | Attribution | Total |
|---|---|---|---|
| LOGOA | 43:15 | 02:53 | 46:08 |
| GUDA (Ours) | 07:18 | 01:36 | 08:54 |
| Wang et al. (2024) | 02:22 | 156:11 | 158:33 |

### D.3. Computational Cost Analysis

We report wall-clock timing for the CIFAR-10 experiment (2,048 queries, single GPU) as a reference; absolute times are implementation-dependent (batch size, hardware, etc.), but relative speedups are expected to hold under different implementations.

**Speedup analysis.** GUDA achieves $\sim100\times$ speedup over LOGOA (02:02 versus 207:47), stemming entirely from preprocessing efficiency: unlearning requires 1/120 as many optimization steps as full retraining (20 epochs versus 2,400 epochs). GUDA and LOGOA share identical query-time costs (1.61s per image) because both evaluate ELBO differences using the same computational procedure. DAS exhibits $7\times$ slower query-time performance (11.62s per image) due to per-query gradient computation.

**Preprocessing versus query-time decomposition.** A key advantage of GUDA is amortizing group-level unlearning (preprocessing, query-independent) across arbitrary query sets. For attribution over $Q$ query images, total time follows $T_{\text{total}} = T_{\text{preproc}} + Q \cdot t_{\text{query}}$, where $t_{\text{query}}$ is per-image attribution cost. GUDA's fixed preprocessing cost becomes increasingly favorable as query set size grows, while maintaining the computational profile of gradient-free attribution methods.

### D.4. Computational Cost Analysis for UnlearnCanvas

We report wall-clock timing for the UnlearnCanvas experiments (Stable Diffusion 1.5, 16-style evaluation, 320 query images) as a reference; absolute times are implementation-dependent, but relative speedups are expected to hold under different implementations.

**Speedup analysis.** GUDA achieves $\sim5.2\times$ speedup over LOGOA (08:54 versus 46:08) and $\sim17.8\times$ speedup over Wang et al. (2024) (08:54 versus 158:33). The speedup over LOGOA stems from preprocessing efficiency: unlearning requires 1/5 as many optimization steps as full retraining (2,000 versus 10,000 steps), yielding $5.9\times$ reduction in preprocessing time. The Wang et al. (2024) baseline exhibits a fundamental scalability bottleneck: attribution time dominates total cost (156:11 out of 158:33) because each query image requires per-image unlearning. Moreover, our evaluation used only 20 images per style (1/20 of the full 400 images per style); scaling to the complete training set would increase the total time for Wang et al. (2024) by $20\times$ to approximately 3,124 hours.

**Preprocessing versus query-time decomposition.** GUDA amortizes group-level unlearning (preprocessing, 07:18) across arbitrary query sets, with per-image attribution cost scaling linearly. For $Q$ query images, total time follows $T_{\text{total}} = T_{\text{preproc}} + Q \cdot t_{\text{query}}$. The Wang et al. (2024) baseline's per-query unlearning cost (29.3 minutes per image) makes it impractical for large-scale attribution, while GUDA maintains consistent speedup across varying query set sizes.

### D.5. Validity of the ELBO Proxy: Comparison with a Likelihood Oracle

The LOGOA attribution score (Eq. (4)) uses $\Delta$ELBO as a surrogate for $\Delta \log p$. Because the ELBO is a lower bound on log-likelihood, the difference between two models decomposes as $\Delta \log p = \Delta\text{ELBO} - \Delta\text{KL}$, where $\Delta\text{KL}$ denotes the change in the KL gap. In principle, unlearning could alter the reverse-process denoising distribution asymmetrically, thereby

*Table 13.* Method comparison under ELBO oracle (main paper) vs. PF-ODE likelihood oracle (5 seeds, aligned batching) on CIFAR-10 (2,048 queries, 10 groups). Best result per column in **bold**; ↑ higher is better.

*(a) ELBO Oracle*

| Method | Top-1↑ | MRR↑ | NDCG@3↑ | Top-3↑ | RBO↑ | Spearman↑ |
|---|---|---|---|---|---|---|
| GUDA (Ours) | **0.727** | **0.798** | **0.677** | **0.475** | **0.423** | 0.265 |
| GUDA w/ ESD | 0.619 | 0.732 | 0.634 | 0.457 | 0.406 | 0.241 |
| CLIPA | 0.662 | 0.755 | 0.646 | 0.462 | 0.412 | 0.246 |
| DAS (Lin et al., 2025) | 0.716 | 0.794 | 0.675 | 0.473 | 0.422 | **0.267** |
| D-TRAK (Zheng et al., 2024) | 0.609 | 0.731 | 0.639 | 0.466 | 0.409 | 0.258 |
| TRAK (Park et al., 2023) | 0.118 | 0.313 | 0.317 | 0.313 | 0.309 | 0.030 |

*(b) Likelihood Oracle*

| Method | Top-1↑ | MRR↑ | NDCG@3↑ | Top-3↑ | RBO↑ | Spearman↑ |
|---|---|---|---|---|---|---|
| GUDA (Ours) | **0.645** | **0.728** | 0.618 | 0.435 | 0.401 | 0.187 |
| GUDA w/ ESD | 0.544 | 0.665 | 0.588 | 0.441 | 0.391 | 0.198 |
| CLIPA | 0.588 | 0.691 | 0.599 | 0.445 | 0.397 | 0.204 |
| DAS (Lin et al., 2025) | 0.639 | 0.726 | **0.630** | 0.459 | **0.408** | 0.236 |
| D-TRAK (Zheng et al., 2024) | 0.566 | 0.687 | 0.610 | **0.462** | 0.403 | **0.260** |
| TRAK (Park et al., 2023) | 0.112 | 0.309 | 0.322 | 0.324 | 0.313 | 0.057 |

changing $\Delta$KL and potentially distorting attribution rankings. We therefore construct an aligned probability-flow ODE (PF-ODE) likelihood oracle to empirically assess how closely $\Delta$ELBO tracks $\Delta \log p$.

**Setup.** We evaluate on CIFAR-10 using the same 2,048 generated queries as in the main experiments. For each query, we estimate $\log p_\theta(x_0)$ by running PF-ODE likelihood estimation (Song et al., 2021) on both the all-class model and each of the 10 LOGO models. A naive single-pass pipeline proved unreliable due to Hutchinson noise and batch-layout mismatches between models, which altered the sign of $\Delta \log p$ on a fraction of query-class pairs. We therefore use an *aligned* evaluation protocol: all models are evaluated on (i) the same contiguous image shard, (ii) the same batch partition, and (iii) the same deterministic Hutchinson seeds. We average over 5 independent seeds and use batch size 64 with 16 GPU shards in parallel (torchdiffeq solver, rtol=$10^{-5}$, atol=$10^{-5}$).

**ELBO–likelihood correlation.** The resulting $\Delta$ELBO and $\Delta \log p$ scores are strongly correlated: overall Pearson = 0.820, Spearman = 0.735. Within-image class-ranking agreement is also high (Top-1 agreement = 0.683, NDCG@3 = 0.939), indicating that $\Delta$ELBO correctly identifies the top contributing class for a large majority of queries. The correlation is not perfect, reflecting both the inherent KL gap and the numerical difficulty of ODE-based likelihood estimation; however, the discrepancy is not large enough to systematically distort head-class identification.

**Method comparison under both oracles.** Table 13 reports all methods evaluated under the ELBO oracle used in the main paper and under the PF-ODE likelihood oracle described above.

Three observations emerge from this comparison. First, Top-1 and MRR generally decrease under the likelihood oracle, confirming that it constitutes a stricter evaluation target while maintaining consistent relative trends. Second, GUDA retains the best Top-1 and MRR under both oracles (0.645 and 0.728, respectively), demonstrating that $\Delta$ELBO is a reliable proxy for identifying the primary contributing group. Third, under the likelihood oracle, DAS and D-TRAK become more competitive on broader ranking metrics (NDCG@3, Top-3, Spearman), suggesting that $\Delta$ELBO is largely sufficient for head identification but may be a coarser approximation for ranking lower-influence groups. Taken together, these results support using $\Delta$ELBO as a practical and empirically grounded surrogate for $\Delta \log p$ in group-wise attribution.

### D.6. Robustness to Noisy Group Partitions

The main experiments assume a clean disjoint partition of training data into groups. To assess robustness when this assumption is violated, we evaluate attribution quality under a noisy group assignment on CIFAR-10.

**Setup.** Starting from the standard 10-class partition (5,000 images per class), we randomly reassign 250 images per class (5% of each group) to other groups. The reassignment follows a balanced no-self derangement (seed 42), so that each group exports and receives exactly 250 images and no image is reassigned to its original class. LOGO counterfactual models are retrained under the same noisy partition, and all methods are evaluated using the identical 2,048-query protocol as the main

*Table 14.* Examples of style-descriptor training and evaluation prompts (16 styles).

| Style | Object | Training prompt | Evaluation prompt |
|---|---|---|---|
| Abstractionism | Architectures | A depiction of Architectures, artistic style featuring vertical, energetic, impasto | An artwork of Architectures, with energetic, impasto, vertical |
| Artist Sketch | Sandwiches | An image showing Sandwiches, artistic style featuring sketchy, soft shading, sketch-like | An artwork of Sandwiches, with soft shading, sketch-like, sketchy |
| Blossom Season | Flame | A detailed view of Flame, artistic style featuring painted effect, vibrant red, impasto texture | An artwork of Flame, with vibrant red, impasto texture, painted effect |
| Blue Blooming | Rabbits | Rabbits as the main subject, artistic style featuring watercolor effect, atmospheric perspective, atmosphere | An artwork of Rabbits, with atmosphere, watercolor effect, atmospheric perspective |
| Bricks | Horses | Horses featured in the scene, artistic style featuring intricate, repeating, symmetrical | An artwork of Horses, with intricate, repeating, symmetrical |
| Byzantine | Fishes | Fishes featured in the scene, artistic style featuring crumpled paper, monochromatic background, subdued | An artwork of Fishes, with monochromatic background, subdued, crumpled paper |
| Cartoon | Statues | A depiction of Statues, artistic style featuring bright colors, overlaid, overlapping layers | An artwork of Statues, with overlaid, bright colors, overlapping layers |
| Cold Warm | Human | A rendering of Human, artistic style featuring splattered paint, watercolor effect, dynamic brushstrokes | An artwork of Human, with watercolor effect, splattered paint, dynamic brushstrokes |
| Color Fantasy | Trees | A depiction of Trees, artistic style featuring psychedelic, wavy lines, chaotic texture | An artwork of Trees, with chaotic texture, wavy lines, psychedelic |
| Comic Etch | Frogs | A rendering of Frogs, artistic style featuring soft pastel, streaked, brushstroke-heavy | An artwork of Frogs, with streaked, brushstroke-heavy, soft pastel |
| Crayon | Horses | Horses featured in the scene, artistic style featuring energetic, whimsical, varied | An artwork of Horses, with whimsical, energetic, varied |
| Crypto Punks | Sandwiches | A rendering of Sandwiches, artistic style featuring intersecting lines, multicolored, interlocking shapes | An artwork of Sandwiches, with multicolored, intersecting lines, interlocking shapes |
| Cubism | Birds | Birds as the main subject, artistic style featuring geometric patterned background, minimalistic, cracked | An artwork of Birds, with minimalistic, cracked, geometric patterned background |
| Dadaism | Cats | Cats as the main subject, artistic style featuring polygonal, warm, warm tones | An artwork of Cats, with warm, warm tones, polygonal |
| Dapple | Birds | A rendering of Birds, artistic style featuring multicolored dots, speckled, circular motifs | An artwork of Birds, with circular motifs, multicolored dots, speckled |
| Defoliation | Bears | Bears featured in the scene, artistic style featuring warm, earthy, mosaic | An artwork of Bears, with mosaic, warm, earthy |

experiments, with the noisy LOGO $\Delta$ELBO as the oracle.

**Results.** Table 16 reports attribution metrics under the noisy partition alongside the clean-to-noisy performance change. All methods degrade moderately relative to the clean setting (roughly 4–6 percentage points on Top-1), and the relative ordering among methods is consistent with the clean-partition results. GUDA retains the best performance on Top-1, MRR, NDCG@3, RBO, and Spearman, with no evidence of catastrophic sensitivity to noisy group definitions.

# E. Limitations

**ELBO as a likelihood surrogate.** Our attribution score uses ELBO differences as a tractable surrogate for log-likelihood differences. While ELBO differences are empirically correlated with $\Delta \log p$ on CIFAR-10 (Appendix D.5), they do not in general preserve log-likelihood ordering: the KL gap between the true posterior and model posterior may change asymmetrically under unlearning, so head identification is reliable but full-ranking correlation may be weaker.

**Approximation quality of the unlearning operator.** The unlearning operator in GUDA-U is a practical approximation of the full ReTrack objective (Eq. (15)): it omits the explicit density-ratio correction and truncates the importance-weighted sum to $K$-nearest neighbors. This introduces controlled bias that we validate empirically against LOGO, but no formal approximation-error bound is provided.

**Disjoint group partition assumption.** The framework assumes a clean, disjoint partition of training data into groups.

*Table 15.* Examples of forget prompts and their corresponding anchor prompts produced by weighted style selection.

| # | Forget style | Forget prompt | Anchor prompt |
|---|---|---|---|
| 1 | Blue Blooming | Towers, artistic style featuring light washes, watercolor, atmospheric perspective. | A depiction of Towers, artistic style featuring contrasting, dynamic shapes, surrealistic. |
| 2 | Abstractionism | Sea featured in the scene, artistic style featuring dynamic forms, energetic, energetic composition. | A depiction of Sea featured in the scene, artistic style featuring grayscale, sketchy, soft shading. |
| 3 | Crypto Punks | A detailed view of Architectures, artistic style featuring intersecting lines, multi-colored, interwoven. | A depiction of A detailed view of Architectures, artistic style featuring soft pastel, brushstroke-heavy, streaked. |
| 4 | Bricks | A composition with Dogs, artistic style featuring repeating, repetitive, abstract design. | A depiction of A composition with Dogs, artistic style featuring digital glitch, circuit board, green and orange. |
| 5 | Defoliation | Sea featured in the scene, artistic style featuring earthy, warm, mosaic. | A depiction of Sea featured in the scene, artistic style featuring loose brushstrokes, low resolution, delicate. |
| 6 | Dadaism | A composition with Dogs, artistic style featuring polygonal, warm, fluid lines. | A depiction of A composition with Dogs, artistic style featuring minimal shading, crumpled paper, monochromatic background. |
| 7 | Blossom Season | A detailed view of Towers, artistic style featuring vibrant red, impasto, contrasting. | A depiction of A detailed view of Towers, artistic style featuring neon colors, digital glitch, futuristic. |

*Table 16.* Attribution performance under a 5% noisy group partition on CIFAR-10 (2,048 queries, 10 groups). Best result per column in **bold**; ↑ higher is better. Sp. denotes Spearman correlation.

*(a) Noisy partition*

| Method | Top-1↑ | MRR↑ | NDCG@3↑ | Top-3↑ | RBO↑ | Spearman↑ |
|---|---|---|---|---|---|---|
| GUDA (Ours) | **0.673** | **0.753** | **0.649** | **0.469** | **0.416** | **0.268** |
| CLIPA | 0.606 | 0.712 | 0.620 | 0.455 | 0.402 | 0.218 |
| DAS (Lin et al., 2025) | 0.666 | 0.751 | 0.645 | 0.458 | 0.412 | 0.244 |
| D-TRAK (Zheng et al., 2024) | 0.581 | 0.700 | 0.621 | 0.468 | 0.405 | 0.245 |

*(b) Δ vs. clean partition*

| Method | Top-1 | MRR | NDCG@3 | Top-3 | RBO | Spearman |
|---|---|---|---|---|---|---|
| GUDA (Ours) | −0.054 | −0.045 | −0.028 | −0.006 | −0.007 | +0.003 |
| CLIPA | −0.056 | −0.043 | −0.026 | −0.007 | −0.010 | −0.028 |
| DAS (Lin et al., 2025) | −0.050 | −0.043 | −0.030 | −0.015 | −0.010 | −0.023 |
| D-TRAK (Zheng et al., 2024) | −0.028 | −0.031 | −0.018 | +0.002 | −0.004 | −0.013 |

Behavior under overlapping or hierarchical group definitions is less understood, though preliminary results in Appendix D.6 suggest robustness to moderate membership noise (5% random reassignment).

**Non-additivity of group scores.** Group attribution scores are not assumed to be additive. Complementarity and redundancy among groups can produce non-additive interactions. More expressive attribution (e.g., Shapley-style extensions to higher-order group effects) is a natural direction for future work.

**Soft reweighting as an alternative estimand.** Soft reweighting of training data defines a different estimand from our hard-removal LOGO target. Studying the relationship between these two notions of group influence is left for future work.

**Scope of experimental evaluation.** Experiments cover artistic style attribution (UnlearnCanvas) and object class attribution (CIFAR-10). Extending GUDA to other group definitions, modalities, or larger-scale models would require additional validation.

