# OpenReview forum: "GUDA: Counterfactual Group-wise Training Data Attribution for Diffusion Models via Unlearning"
_ICML.cc/2026/Conference — ICML 2026 regular_

### Official Review · Reviewer_LAvR · 2026-03-03

**Soundness:** 3
**Presentation:** 4
**Significance:** 3
**Originality:** 3
**Overall Recommendation:** 5
**Confidence:** 3

**Summary:**

The authors propose a new method called Group Unlearning-based Data Attribution (GUDA), which aims to analyze the counterfactual scenario where a group of data was not included in the training set. Contrary to the baseline Leave-One-Group-Out (LOGO) retraining, this is much more efficient.

**Compliance With Llm Reviewing Policy:**

Affirmed.

**Key Questions For Authors:**

1) How does your work contrast your work from [1] aside from being for images instead of music? I think it would be good to mention this paper at least and contrast a bit with it.

[1] Choi, Woosung, et al. "Large-scale training data attribution for music generative models via unlearning." arXiv preprint arXiv:2506.18312 (2025).

**Limitations:**

Not really.

**Strengths And Weaknesses:**

Pros:
+ Being able to fight, e.g. copyright infringements, is an important challenge in generative modelling, and this seems like a useful tool for it
+ well-written
+ figure 1 explains the whole paper quite well
+ beats related work on all metrics aside from Spearman on CIFAR-10 while being much faster
+ qualitative results from Figure 2 seem convincing as well

Cons:
- the technical novelty is okay, but not exceptional

---

> ### Author Rebuttal · Authors · 2026-03-31
>
> We thank the Reviewer for their careful and thoughtful evaluation. We are greatly encouraged that the Reviewer recognized GUDA as a potentially useful tool for addressing an important challenge in generative modeling, and appreciated the clarity of the framework conveyed through Figure 1 as well as the convincing qualitative results in Figure 2. We believe these comments accurately reflect the practicality, efficiency, and interpretability that we prioritized in the design of GUDA.
>
> **[Q1] Relationship with the related paper**
>
> We will explicitly cite the paper by Choi et al. in the rebuttal and revised manuscript, and provide a clearer discussion of the relationship. That said, the differences between the two works extend well beyond simply "handling images instead of music." Choi et al. apply unlearning-based attribution to large-scale text-to-music diffusion models to identify specific training data points that contributed to a given output, that is, they primarily address instance-level attribution. In contrast, the central focus of our work lies in the formulation of a group-removal counterfactual: "how would the model's explanatory power change if a specific group had been absent during training?" To this end, we define LOGO as a conceptual oracle, construct the counterfactual model itself, the target of approximation, via unlearning, and measure group-wise influence through the ELBO difference between the full model and the counterfactual model. Thus, the novelty of our work does not reside in merely repurposing unlearning for attribution, but rather in explicitly defining a group-wise counterfactual estimand and validating it through LOGO fidelity.
>
> From this perspective, the methodologically closer related work is in fact Wang et al. (2024). Indeed, in our manuscript we already clarify the differences from Wang et al. in the Related Work section along two axes: (i) our estimation target is the group-removal counterfactual rather than instance-level attribution, and (ii) we unlearn the target group to approximate the counterfactual model, rather than unlearning a synthesized image to obtain loss-based ranking signals. Furthermore, we include Wang et al. as a direct baseline in our UnlearnCanvas experiments, where GUDA demonstrates superiority in both agreement with the LOGOA head and computational efficiency. Accordingly, we position Choi et al. as an important related work demonstrating that unlearning-based attribution can be extended to another modality (music), while our most direct baseline remains Wang et al., against which we have already conducted experimental comparisons.
>
> We also note that music is a domain where attribution naturally connects to artist-level recognition and fair compensation, making it a particularly fitting application for unlearning-based attribution. In this sense, Choi et al. is complementary rather than competitive with our work. We believe that both directions, group-wise counterfactual attribution in the vision domain and large-scale instance-level attribution in the music domain, are mutually important for advancing transparency and equitable value distribution in generative models.

---

> > ### Author Rebuttal · Reviewer_LAvR · 2026-04-01
> >
> > I am happy with the paper and from my perspective it could be published at ICML

---

### Official Review · Reviewer_Ayn7 · 2026-03-05

**Soundness:** 2
**Presentation:** 3
**Significance:** 2
**Originality:** 2
**Overall Recommendation:** 4
**Confidence:** 5

**Summary:**

This paper introduces GUDA, a framework for group-wise training-data attribution in diffusion models that approximates expensive leave-one-group-out (LOGO) retraining with machine unlearning from a shared full-data model. Instead of asking which individual examples mattered, the paper asks which groups of data, e.g., CIFAR-10 classes or artistic styles, most influence a generated output, and measures that influence by the drop in a likelihood surrogate (ELBO) between the full model and a counterfactual model with one group removed. The method combines a preservation term to maintain performance on retained data with a setting-specific forget term: ReTrack-style redirection for unconditional generation and a conditional anchor-based redirection for text-to-image style attribution. Empirically, the paper reports that GUDA better matches the LOGO oracle than similarity-based and instance-level attribution baselines, while being far more computationally feasible than retraining one model per group.

**Compliance With Llm Reviewing Policy:**

Affirmed.

**Final Justification:**

The rebuttal provides a helpful additional experiment that addresses my ELBO proxy concern. It empirically demonstrates strong alignment with log-likelihood differences, which increases confidence about the validity and practicality of the method. However, the proposal still lacks deeper theoretical justification and leaves open questions around robustness and group interactions. Overall, I am relatively convinced and will raise my score from "weak reject" to "weak accept".

**Key Questions For Authors:**

1. **Hard vs. soft interventions.** The method seems to model group influence through hard removal/retention of a group. Why is this the right intervention, and could a soft reweighting formulation be more stable or more informative?
2. **Group definition and robustness.** How exactly are groups defined, and how sensitive are the results to small changes in membership? For example, if a single instance is added to or removed from a group, do the approximation guarantees or attribution rankings remain stable?
3. **Additivity of group effects.** Does the method assume or empirically observe any form of additivity across groups, or can interactions between groups make group-wise attributions strongly non-additive?
4. **Assumptions on the data partition.** Are there any assumptions on how the training data is split into groups, e.g., disjointness, semantic coherence, balanced size, or shared distributional support? If so, how do violations of these assumptions affect the method?

**Limitations:**

Yes.

**Strengths And Weaknesses:**

### 1. Soundness.
- **W1.1 – ELBO as a tractable proxy.** Replacing the log-likelihood difference with an ELBO difference (Eq. 4) silently assumes the KL gap $G_\theta(x_0,c)=\log p_\theta(x_0\mid c)-\mathrm{ELBO}(x_0\mid c;\theta)$ is approximately the same for the full and leave-one-group-out models. Concretely, the true counterfactual decomposes as  $\Delta_k = \bigl[\mathrm{ELBO}^{\mathrm{full}}-\mathrm{ELBO}^{-k}\bigr]+\bigl[G_{\theta^{\mathrm{full}}}-G_{\theta^{-k}}\bigr],$  so ELBO differences equal likelihood differences only when the gap difference is negligible. It is not clear why this would be guaranteed under unlearning, which could degrade the reverse process at certain timesteps, misaligning the learned denoising distribution with the true posterior, hence changing the KL term asymmetrically. The authors acknowledge ELBO "does not guarantee preservation of likelihood ordering", but do not discuss when or how badly the gap difference could distort (maybe even flip the sign of) attribution scores. Empirical evidence that the gap remains stable under unlearning—or an analysis of when it does not—seems necessary to support this approach.
- **W1.2 – Claims and construction.** The causal language feels overstated: this is closer to estimating the effect of group-level data interventions than establishing full counterfactual causal influence in the SCM sense. On groups, there is but limited discussion of how sensitive the method is to group definition or membership changes, even though group construction seems central to the guarantees and interpretation.

### 2. Significance.
- **S2.1 – Problem definition.** The paper addresses an important problem for generative models: identifying which groups of training data most influence outputs. The prposed approach is potentially more scalable than retraining a separate model for each removed group, which is practically meaningful. The experiments suggest gains over existing baselines (gradient-based or instance-level unlearning based), and makes the case for GUDA's practical significance.

### 3. Originality.
- **W3.1 – Prior work positioning is incomplete.** GUDA appears closely related to prior work on training-data attribution, including Datamodels, group-based influence functions, and unlearning based attributions [1, 2, 3]. Prior lines of work study the counterfactual effects of training data on model behaviour, and some already address group-level attribution explicitly. As written, the paper does not clearly isolate whether its main novelty lies in the group formulation, the use of machine unlearning to approximate leave-one-group-out retraining, or the diffusion-specific redirection design. A fuller discussion of existing literature would make the originality claim more convincing.
- **W3.2 – Methodological novelty.** The unconditional setting seems close to being a direct application of ReTrack-style redirection, which makes the incremental novelty there somewhat unclear. The conditional variant appears more original, but the paper should better clarify the technical contribution beyond adapting ReTrack with CLIP-weighted or anchor-based redirection. At present, the reviewer is left unsure whether the main novelty is methodological or primarily an application-specific extension.

### 4. Presentation.
- **S4.1 – Scope and framing.** The paper clearly differentiates group-level attribution from instance-level attribution, especially in terms of scalability and nonlinear interactions. The decomposition into preservation and forgetting terms is intuitive and makes the method readable; differences in implementation between unconditional and conditional settings is well documented.
- **W4.2 – Clarity.** Some core concepts need sharper explanation, especially what the LOGO oracle exactly means in practice and how it differs from simpler baselines such as repeated retraining or cross-validation-style setups. The paper could be more explicit about what kind of "counterfactual" it is estimating, to avoid confusion between model intervention, policy-level intervention, and unit-level causal reasoning.

**References.**
[1] Ilyas, Andrew, et al. "Datamodels: Predicting Predictions from Training Data." Proceedings of the 39th International Conference on Machine Learning. 2022.
[2] Fan, Chongyu, et al. "SalUn: Empowering Machine Unlearning via Gradient-based Weight Saliency in Both Image Classification and Generation." The Twelfth International Conference on Learning Representations.
[3] Choi, Ching Lam, Alexandre Duplessis, and Serge Belongie. "Unlearning-based Neural Interpretations." The Thirteenth International Conference on Learning Representations.

---

> ### Author Rebuttal · Authors · 2026-03-31
>
> We thank the reviewer for the careful and constructive assessment. We are encouraged that the reviewer recognizes the importance of the group-wise attribution problem and the practical value of approximating LOGO retraining with unlearning. We agree that two points require sharper clarification: first, that our attribution score uses ELBO as a tractable surrogate rather than exact likelihood, and second, that our method targets a group-removal counterfactual at the model level, not causality in the full SCM sense. In the rebuttal and revision, we focus on clarifying these points, in addition to better positioning our methodological novelty relative to prior work.
>
> **[W1.1] Validity of the ELBO proxy**
>
> We agree that ELBO differences do not in general coincide with log-likelihood differences, and that unlearning could change the KL gap asymmetrically, potentially distorting rankings or even flipping signs.
>
> **Additional experiment.** We therefore constructed a probability-flow ODE log-likelihood oracle on CIFAR-10 (5 seeds, aligned batching) and compared ΔELBO against Δlog-likelihood across all 2,048 queries.
>
> ΔELBO and Δlog-likelihood are strongly correlated (Pearson = 0.820, Spearman = 0.735), with high within-image ranking agreement (Top-1 = 0.683, NDCG@3 = 0.939). The correlation is not perfect, which reflects both the KL-gap asymmetry noted by the reviewer and the inherent difficulty of obtaining precise log-likelihood estimates for diffusion models (exact computation requires ODE integration with variance reduction, costing orders of magnitude more than ELBO). Nonetheless, switching to the log-likelihood oracle does not substantially change the overall trends: all methods degrade uniformly, and GUDA still achieves the best Top-1 and MRR (0.645 / 0.728 vs. DAS 0.639 / 0.726 and CLIPA 0.588 / 0.691). On broader ranking metrics (NDCG@3, Spearman), DAS and D-TRAK become more competitive, suggesting that ELBO is sufficient for the main trends, while ranking lower-influence groups is more sensitive to the surrogate.
>
> In the revision, we will report these results and position ELBO explicitly as a practical computable surrogate, while discussing when KL-gap asymmetry may affect rankings.
>
> **[W1.2 / Q1 / Q2 / Q4] Causal language and sensitivity to group definition**
>
> The "counterfactual" in our work refers to a model-level operational counterfactual, namely how the model's explanatory power changes when group k is removed from training, rather than a unit-level causal counterfactual in the SCM sense. We agree the causal language was too strong, and in the revision we will uniformly adopt "group-removal counterfactual effect."
>
> We also agree that sensitivity to group definition is important. Although the current version assumes disjoint groups, as shown in our response to wx3h [Q3], we evaluated GUDA under a noisy partition setting. GUDA maintained the best performance, suggesting that it is not catastrophically fragile to moderate membership noise and may tolerate other types of group noise as well. Ambiguous group definitions, overlapping-group settings, and finer-grained membership perturbations remain open, and we will state these limitations explicitly.
>
> Regarding soft reweighting, we adopt hard removal because our target applications naturally ask "what if this group had been absent?", which directly matches the LOGO oracle. Soft reweighting defines a different estimand and is an interesting future direction.
>
> **[W3.1 / W3.2] Positioning of novelty relative to prior work**
>
> We agree that the original draft did not clearly isolate where the novelty lies. In the revision, we will clarify our novelty along three distinct axes: (i) **estimand**, an explicit per-generation group-removal counterfactual target (LOGOA) for diffusion models, with LOGO retraining as oracle; (ii) **approximation**, group unlearning as a practical approximation to that counterfactual, with fidelity evaluated directly against the LOGO oracle; and (iii) **conditional design**, a text-to-image attribution mechanism that addresses prompt shift under group removal. This framing, rather than a standalone new redirection method, is the main distinction from Datamodels, group-based influence methods, and prior unlearning-based attribution. We also agree that the unconditional instantiation is close to applying ReTrack within this framework. In contrast, the conditional setting is not a naive extension, because the forget condition lies outside the retain-only support, which motivates our anchor-based redirection; its effectiveness is further supported by the ablations reported in our response to wx3h [Q2].
>
> **[Q3] Additivity of group effects**
>
> We do not assume additivity of group effects. In general, group scores need not sum to the total effect, and complementarity or redundancy can make them strongly non-additive. Modeling such interactions more explicitly, for example via Shapley-style extensions, is a natural future direction.

---

> > ### Author Rebuttal · Reviewer_Ayn7 · 2026-04-04
> >
> > The rebuttal provides a helpful additional experiment that addresses my ELBO proxy concern. It empirically demonstrates strong alignment with log-likelihood differences, which increases confidence about the validity and practicality of the method. However, the proposal still lacks deeper theoretical justification and leaves open questions around robustness and group interactions. Overall, I am relatively convinced and will raise my score from "weak reject" to "weak accept".

---

### Official Review · Reviewer_wx3h · 2026-03-09

**Soundness:** 3
**Presentation:** 3
**Significance:** 3
**Originality:** 3
**Overall Recommendation:** 5
**Confidence:** 5

**Summary:**

This paper studies group-wise data attribution for diffusion models.
Rather than assigning influence to individual training examples and aggregating them afterward, the paper directly asks a group-removal counterfactual question: how much a training group contributes to a generated sample if that group were removed from training.
To formalize this, the paper defines a leave-one-group-out attribution target based on the ELBO difference between the full model and a counterfactual model trained without the target group, using LOGO retraining as the conceptual oracle.
Since retraining one model per group is prohibitively expensive, the paper proposes GUDA, which approximates these counterfactual models via machine unlearning from the full model.
The method is instantiated for both unconditional diffusion and conditional text-to-image diffusion, and evaluated on CIFAR-10 and UnlearnCanvas.
Overall, the paper shows that GUDA can better match the LOGO oracle than semantic-similarity and instance-level attribution baselines, while being substantially more efficient than explicit LOGO retraining.

**Compliance With Llm Reviewing Policy:**

Affirmed.

**Final Justification:**

My concerns have been adequately addressed.

**Key Questions For Authors:**

1. The paper uses ELBO differences as the operational attribution score, while noting that ELBO may not preserve likelihood ordering. Can the authors discuss more concretely how often this mismatch matters in practice, and whether attribution rankings remain stable under this surrogate?

2. In the conditional text-to-image setting, can the authors provide more justification or ablation for the anchor/style construction choices? This would help clarify how robust the method is to these design decisions.

3. The current setup assumes disjoint groups. How do the authors expect the method to behave when groups overlap or when the grouping is noisy and less curated?

4. To what extent does GUDA depend on the specific unlearning procedure used here? Do the authors expect the framework to transfer to other unlearning recipes with similar attribution fidelity?

**Limitations:**

yes

**Strengths And Weaknesses:**

1. A major strength of the paper is that it formulates group-wise attribution for diffusion models in a principled way, instead of relying on instance-level attribution followed by aggregation. This is better aligned with practical questions such as understanding the contribution of a class or style to a generated sample, and I think this shift in formulation is meaningful and well motivated. The paper also clearly explains why group effects are not trivially reducible to sums of instance-level effects.

2. Another strength is that the paper proposes a practical approximation to an otherwise expensive counterfactual target. Using LOGO retraining as the conceptual oracle and unlearning as the approximation is a sensible design, and the empirical comparisons against the LOGO oracle make the evaluation more convincing than if the paper had only compared to weaker baselines. The reported results on CIFAR-10 and UnlearnCanvas suggest that the method is effective for the settings studied, while being much more efficient than explicit retraining.

3. My main reservation is that the paper still has some methodological and scope limitations that prevent a stronger recommendation. In particular, the use of ELBO as a surrogate introduces a gap between the ideal attribution target and the practical score, and the paper itself acknowledges this limitation. In addition, the conditional variant feels somewhat more heuristic than the unconditional case, and the experiments are still limited to relatively clean, disjoint group structures. Overall, I find the paper technically solid and interesting, but the current version feels somewhat narrow in scope and not yet fully convincing in all settings.

---

> ### Author Rebuttal · Authors · 2026-03-31
>
> We thank the reviewer for the constructive evaluation. We are encouraged that the reviewer finds the group-wise formulation meaningful, the LOGO-based evaluation convincing, and the approach technically solid. We focus on clarifying the main limitations raised, namely the stability of the ELBO surrogate, design choices in the conditional variant, and robustness beyond disjoint-group settings.
>
> **[W1 / Q1] ELBO surrogate and ranking stability**
>
> Due to space constraints, we provide an empirical analysis of this point in our reply to Ayn7 [W1.1], who raised a closely related question. In brief, we constructed a probability-flow ODE log-likelihood oracle on CIFAR-10 and found that ΔELBO and Δlog-likelihood exhibit strong correlation (Pearson = 0.820, Spearman = 0.735). While KL gap asymmetry is non-negligible and uniformly degrades all methods under the stricter oracle, GUDA retains the best Top-1 and MRR among compared methods, confirming that our conclusions, particularly regarding head identification, remain robust under the log-likelihood ground truth.
>
> **[W2 / Q2 / Q3] Anchor design for the conditional variant / Robustness to noisy group assignment**
>
> **Conditional variant (GUDA-C)**
>
> We acknowledge GUDA-C involves additional design choices. However, group removal shifts not only the image distribution but also the prompt distribution (Sec. 4.2.2), making direct application of ReTrack difficult. Anchor-based redirection is a feasible approximation to this challenge, and its validity is confirmed by alignment with the LOGO oracle (Table 2: Top-1 45.6% vs. CLIPA 33.8%).
>
> **Additional experiments: Anchor construction ablation (Q2)**
>
> To validate the robustness of the anchor design choices in GUDA-C, we conducted three ablation studies on UnlearnCanvas under the same 16-style evaluation protocol as Table 2.
>
> 1. **Sampling strategy (weighted vs. uniform):** We compared the proposed CLIP-similarity-weighted style selection against uniform random selection over retain styles. All other settings are identical.
>
> | Sampling | Top-1 | MRR | NDCG@3 | Top-3 | RBO | Spearman |
> |---|---|---|---|---|---|---|
> | Weighted (proposed) | 0.456 | 0.582 | 0.734 | 0.405 | 0.446 | 0.239 |
> | Uniform | 0.447 | 0.552 | 0.680 | 0.332 | 0.410 | 0.104 |
>
> Weighted selection outperforms uniform across all metrics, confirming the value of semantically proximate redirection.
>
> 2. **Hyperparameter sensitivity:** We varied τ ∈ {0.5, 1.0, 2.0, 4.0} and the number of style descriptors k ∈ {1, 3, 5}. All configurations outperformed the uniform baseline with stable performance; k=5 yielded the best results and will be reflected in the revision.
>
> 3. **Anchor construction strategy comparison (AWSS vs. style removal):** We compared the proposed anchor redirection (AWSS) against a baseline that simply removes style descriptors from the forget prompt, corresponding to the neutral-target approach in existing unlearning methods.
>
> | Method | Top-1 | MRR | NDCG@3 | Top-3 | RBO | Spearman |
> |---|---|---|---|---|---|---|
> | AWSS (proposed) | 0.456 | 0.582 | 0.734 | 0.405 | 0.446 | 0.239 |
> | Style removed | 0.400 | 0.526 | 0.671 | 0.331 | 0.409 | 0.118 |
>
> AWSS outperforms simple style removal on all metrics, indicating that merely stripping style information is insufficient to approximate the LOGO counterfactual.
>
> **Additional experiments: Noisy group partition setting (Q3)**
>
> To directly test robustness to imperfect group assignments, we constructed a CIFAR-10 variant where ~5% of training samples (250 per class) are randomly assigned to different groups. LOGO oracles are retrained under the same noisy partition; the query set and metrics are identical to the main experiment.
>
> | Method | Top-1 | MRR | NDCG@3 | Top-3 | RBO | Spearman |
> |---|---|---|---|---|---|---|
> | GUDA (Ours) | 0.673 | 0.753 | 0.649 | 0.469 | 0.416 | 0.268 |
> | CLIPA | 0.606 | 0.712 | 0.620 | 0.455 | 0.402 | 0.218 |
> | DAS | 0.666 | 0.751 | 0.645 | 0.458 | 0.412 | 0.244 |
>
> GUDA retains the best performance, with moderate degradation from the clean setting (~5 percentage points on Top-1). The relative ordering among methods is consistent with the clean setting, and we find no evidence that GUDA is fragile to noisy group definitions. This also suggests potential robustness to other types of membership noise, which we leave for future investigation.
>
> **[Q4] Dependence on the specific unlearning procedure**
>
> The essence of GUDA lies in using unlearning as an efficient approximation to the LOGO counterfactual; the specific unlearning method is interchangeable. In CIFAR-10 experiments (Table 1), ReTrack outperforms ESD, reflecting its theoretical grounding as an unbiased reformulation of the retain-only objective. The choice of unlearning method directly affects attribution fidelity, and methods with stronger alignment to the LOGO counterfactual yield better results. Future unlearning methods can be incorporated within GUDA, with fidelity verifiable against the LOGO oracle.

---

> > ### Author Rebuttal · Reviewer_wx3h · 2026-04-03
> >
> > Thanks, I raised my score.

---

### Decision · Program_Chairs · 2026-04-30

**Decision:**

Accept (regular)

**Comment:**

This paper presents GUDA, a framework for group-wise training data attribution in diffusion models. It aims to investigate how much a generated sample depends on a training group if that group had been removed from training. To answer this question, the authors used machine unlearning to approximate the corresponding counterfactual models, and instantiates the approach for both unconditional and conditional diffusion settings.

Reviewers found the problem formulation meaningful and well motivated. In particular, the use of LOGO retraining as a conceptual oracle, together with unlearning as an efficient approximation, is a very interesting design. Also, the technical approach is clearly presented, and the empirical performance of the proposed method is quite promising, which outperforms relevant baselines in most settings. Reviewers also raised a few concerns, e.g., the scope of experiments might be limited to relatively clean disjoint group structures, positioning of prior work to clarify the novelty, etc. These concerns were mostly addressed by the authors’ rebuttal.

Overall, this is a solid and timely contribution on an important problem, and I encourage the authors to incorporate the additional clarifications, discussion of limitations, and new experimental results into the final version.